# The *Drosophila* Individual Activity Monitoring and Detection System (DIAMonDS)

Ki-Hyeon Seong[1,2†‡*], Taishi Matsumura[3†], Yuko Shimada-Niwa[2,4], Ryusuke Niwa[2,4], Siu Kang[2,3†‡*]

[1]RIKEN Cluster for Pioneering Research, RIKEN Tsukuba Institute, Tsukuba, Japan; [2]AMED-CREST, AMED, Tokyo, Japan; [3]Graduate School of Science and Engineering, Yamagata University, Jonan, Yonezawa, Japan; [4]Life Science Center for Survival Dynamics, University of Tsukuba, Tsukuba, Japan

**Abstract** Here, we have developed DIAMonDS (*Drosophila* Individual Activity Monitoring and Detection System) comprising time-lapse imaging by a charge-coupled device (CCD) flatbed scanner and Sapphire, a novel algorithm and web application. DIAMonDS automatically and sequentially identified the transition time points of multiple life cycle events such as pupariation, eclosion, and death in individual flies at high temporal resolution and on a large scale. DIAMonDS performed simultaneous multiple scans to measure individual deaths (≤1152 flies per scanner) and pupariation and eclosion timings (≤288 flies per scanner) under various chemical exposures, environmental conditions, and genetic backgrounds. DIAMonDS correctly identified 74–85% of the pupariation and eclosion events and ~ 92% of the death events within ± 10 scanning frames. This system is a powerful tool for studying the influences of genetic and environmental factors on fruit flies and efficient, high-throughput genetic and chemical screening in drug discovery.

**\*For correspondence:**
ki-hyeon.seong@riken.jp (K-HS);
siu@yz.yamagata-u.ac.jp (SK)

†These authors contributed equally to this work
‡These authors jointly supervised this work

**Competing interests:** The authors declare that no competing interests exist.

## Introduction

Animal life develops through a sequence of characteristic events and stages. Embryogenesis starts with the fertilization of an egg. Thence, the embryo transitions to the juvenile stage at a certain time point. This stage persists for a certain period, during which the animal grows and develops into the adult stage in which it reproduces, senesces, and dies. The length of each stage and the timing of each developmental transition are determined by complex interactions between genetic and environmental factors. Thus, developmental timing may be defined as a phenotype. Accurate determination of developmental timing helps elucidate the molecular and physiological bases of biological events and may uncover new factors regulating development and lifespan.

Nevertheless, it is difficult to establish the precise time points of the developmental transitions in many animals including humans. One reason is that there may be no clear boundaries between stages throughout the entire lifetime of the organism. Consequently, it has been impractical to use developmental timing as a phenotype for investigations in developmental biology. However, holometabolous insects such as the fruit fly *Drosophila melanogaster* are important exceptions to this rule. These animals pass through four life stages, namely embryo, larva, pupa, and adult. Transitions between these stages are accompanied by drastic morphological events and behavioral changes including hatching, pupariation, eclosion, and death. Therefore, the time point of each developmental transition can be precisely determined. *Drosophila has been extensively* studied in the fields of genetics and developmental biology. Accurate tracking and recording of its life cycle could significantly promote the understanding of various aspects of biology, agriculture, and medicine.

Noteworthy, no recent improvements have been made in measuring the timing of the developmental transitions in *Drosophila*. Until now, the timing of each developmental stage was manually determined by counting the number of flies at each stage in each vial over 2, 6, 12, or 24 hr (*Buhler et al., 2018*; *Demay et al., 2014*; *Külshammer et al., 2015*; *Nikhil et al., 2016*; *Yun et al., 2017*); however, this technique has certain limitations. Increasing the temporal resolution is difficult as it would require intensive labor, preventing the detection of subtle changes in transition timing for each event. Moreover, manual counting is neither practicable for large-scale/high-throughput screening nor feasible for identifying multiple phenotypes in individual flies in transition events. In contrast, it does help elucidate the associations between developmental stages and the genetic and environmental factors affecting them. Recent studies reported the use of video cameras to measure developmental timing (*Hironaka et al., 2019*). However, this method is labor-intensive, requires long analytical periods, and is unsuitable for high-throughput analysis.

Here, we present a new scalable method that automatically determines multiple transition time points, such as pupariation, eclosion, and death, of individual fruit flies implementing a basic flatbed CCD scanner. We placed a single fly in each well of a 96- or 384-well microplate, acquired time-lapse images, and analyzed the output with our novel algorithm Sapphire. This system performs automatic developmental analyses at high temporal resolution and could be used in high-throughput gene and chemical screening and analyses of the effects of genetic and environmental factors on development.

## Results

### Design of the *Drosophila* Individual Activity Monitoring and Detection System (DIAMonDS)

*Drosophila* develops through four developmental stages (embryo, larva, pupa, and adult), subsequently finishing its own life. The static phases (embryo, pupa, and death) alternate with the dynamic phases (larva and adult; *Figure 1A*). Single-image processing between continuous images distinguishes both phases: dynamic phases are detected as positive signals while static phases simultaneously present no signal (*Figure 1B*). Therefore, we can precisely identify the transition points between phases by monitoring the static and dynamic status and separately detecting fly phases on all plates captured simultaneously in the same image.

DIAMonDS consists of an automated time-lapse imaging system and our novel image analysis software named 'Sapphire' (*Figure 1B*). We used a combination of a flatbed CCD scanner and VueScan software (https://www.hamrick.com), which enables multiple scanner units to capture images continuously at certain intervals (*Smith et al., 2014*). To identify the phase change time points for each fly event, single flies are inserted into each well of a 96- or 384-well microplate containing suitable fly media. Up to three microplates are then set on the scanner surface, and time-lapse images are acquired at appropriate intervals until the fly event is completed. A single scanner can monitor 288 or 1152 individuals in three 96-well or 384-well microplates, respectively for DIAMonDS. Because the well-size of 384-well microplate is too small to succeed in normal development and lifespan, we only use the 384-well microplate for measuring adult survivorships in the short term (within about 2 weeks) such as stress or drug resistance assays. Sapphire then automatically analyzes and detects the transition points of pupariation, eclosion, and death in the newly acquired time-lapse images (*Figure 1B,C*). Our system can simultaneously monitor multiple scanners with a single personal computer.

As positional bias and fluctuations in environmental factors such as ambient temperature and relative humidity (RH) might markedly reduce the reliability of our system, all experiments were conducted in a plant growth chamber (LPH-410NS) with automatic temperature and humidity regulation and additional USB fans (Appendix 6). The scan surface temperature was continuously recorded with button-sized temperature data loggers (NK Labs LLC, Cambridge, MA, USA). Temperatures widely varied among locations under external illumination on/off conditions possibly because of irregular and uneven irradiation. Thus, we acquired time-lapse images without external illumination to maintain a steady temperature (*Figure 1—figure supplement 1*).

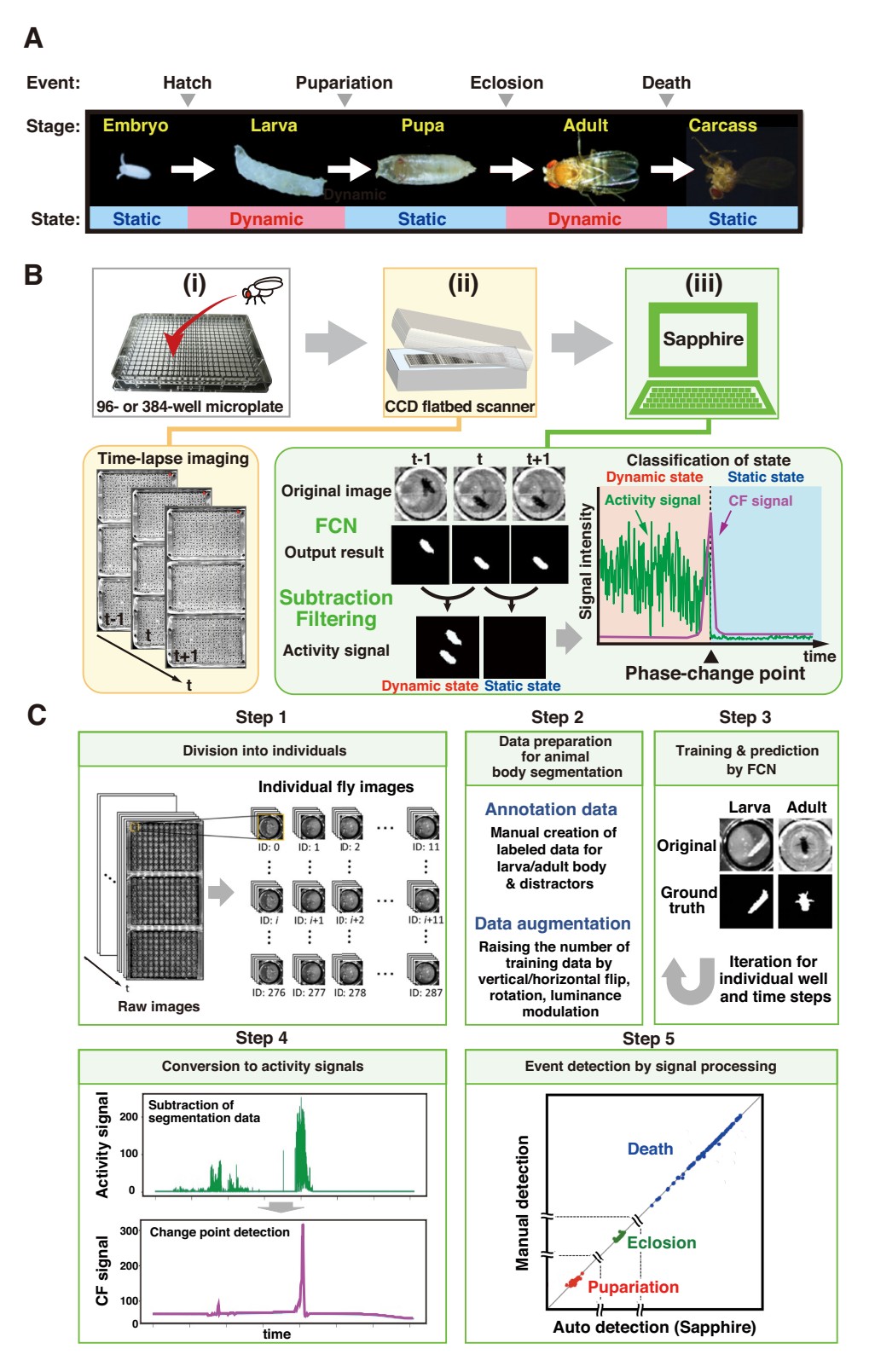

**Figure 1.** Overview of DIAMonDS. (**A**) Diagram of developmental stages and activity states in *Drosophila melanogaster*. (**B**) Schematic representation of the DIAMonDS procedure. DIAMonDS consists of: (i) microplate preparation; (ii) time-lapse imaging with CCD scanner; and (iii) data analysis by Sapphire. In step (iii), Sapphire calculates activity signal (green line) intensity via animal body detection from FCN images and subtraction processing of every two consecutive images and then determines the CF signal (purple line) from the activity signal. (**C**) Flowchart of Sapphire. Algorithm includes

*Figure 1 continued on next page*

*Figure 1 continued*

extraction of individual animals from population images (Step 1), training data preparation and augmentation (Step 2), training through data and animal body segmentation (Step 3), segmentation data signaling by subtracting labeled data and transition point detection algorithm (Step 4), event detection, signal processing, and visualization (Step 5).

The online version of this article includes the following source data and figure supplement(s) for figure 1:

**Figure supplement 1.** Scanning surface temperature distribution.
**Figure supplement 1—source data 1.** Scanning surface temperature.
**Figure supplement 2.** Schematic representation of process of Sapphire and architecture of FCN.
**Figure supplement 3.** Sapphire user interface.

## DIAMonDS software: Sapphire

Sapphire automatically determines the static-to-dynamic and dynamic-to-static phase changes for all flies according to the time-lapse images acquired by the aforementioned scanner system. The following four processes were implemented in Sapphire to enable automatic life event detection based on individual *Drosophila* images (*Figure 1C*):

1. Images of single animals were segregated by image processing (*Figure 1C*, Step 1), enabling the system to readily target individual flies.
2. Semantic segmentation was performed to capture each insect in each well (*Figure 1C*, Step 2). Recently, artificial intelligence techniques have substantially improved both image classification (*Goodfellow et al., 2016*) and segmentation (*Badrinarayanan et al., 2017*; *Ronneberger et al., 2015*). Here, we designed a fully convolutional network (FCN) specifying larval and adult segmentation. The FCN has encoder-decoder architecture (*Figure 1—figure supplement 2*) comprising three blocks, each including a convolution layer with a 3 × 3 filter, up/down sampling layer with a 2 × 2 filter, and dropout layer with a 25% ratio. The encoder and decoder parts were mutually connected by two convolution layers with 3 × 3 filters. The convolution layers in the encoder and decoder were fitted with rectifier linear units. The inputs to the layers were applied by batch normalization. The output layer was also a convolution layer and included reshaping and softmax functions.

The annotation data were manually created for larval and adult *Drosophila* and increased by general data augmentation techniques such as vertical and horizontal flips, rotation, and luminance modulation (Materials and methods).

After training with augmented data, the segmentation inference was calculated as a probability. If it was > 0.5, the system regarded it as the target region (animal body). If it was < 0.5, the system treated it as background. Consequently, the system obtained binary images wherein the animal body corresponded to one and the background was described as 0. Semantic segmentation was applied to all individuals and every sequential population image.

All trainings and inferences were performed on a Linux PC (Ubuntu) with four GPUs (GTX 1080Ti). All scripts were written in Python using the deep learning libraries keras (v. 2.0.9) and tensorflow-gpu (v. 1.4.0).

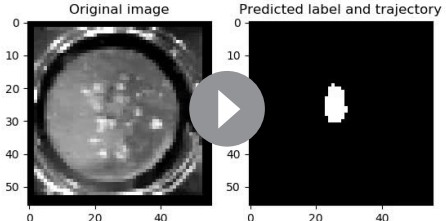

frame=0, #well=94

Original image Predicted label and trajectory

**Video 1.** The estimated activity history of a single fly by Sapphire.
https://elifesciences.org/articles/58630#video1

1. The system converted the labeled data to time series data by subtracting every two consecutive labeled images (*Figure 1C*, Step 3). After signaling, the system applied the ChangeFinder (CF) algorithm developed to detect turning points (*Takeuchi and Yamanishi, 2006*) and evaluated total individual animal activity from the central gravity distance of the segmented area between two consecutive images (*Video 1*).
2. The system automatically determined life event transitions with single-animal resolutions from the CF signals (*Figure 1C*, Step

4). Life event transitions were estimated based on the transition points from dynamic to static or vice-versa. These were designated the maximum points in the CF signals. Death and pupariation corresponded to maximum CF points reflecting dynamic-to-static phase changes. In contrast, eclosion was characterized as a maximum CF point corresponding to a static-to-dynamic change. After transition point determination, the system summarized the automatic detection results using various visualization styles (*Figure 1C*, Step five and *Figure 1—figure supplement 3*). All algorithms and visualization tools were user-friendly web applications developed with the Dash framework in Python (*Figure 1—figure supplement 3*).

## Pupariation and eclosion timing detection by DIAMonDS

Apparent dynamic-to-static and vice-versa phase changes occur at pupariation and eclosion (*Figure 2A*). We attempted to detect individual fly pupariation and eclosion with DIAMonDS to validate it. Newly hatched first-instar (L1) larvae were placed into 96-well microplates containing 100 μL well$^{-1}$ standard fly media (*Figure 2—figure supplement 1*). Three microplates were fixed to the scanner surface, and the scanner was placed inverted in the incubator to prevent fly media leakage. The scanner was powered on, and time-lapse scanning was performed with VueScan (Materials and methods) until all flies eclosed. Thence, time-lapse images were analyzed with Sapphire (*Figure 2A, B*).

The L1 larvae dove into the media until mid-L3 (~2 d). As time-lapse scanning could not detect larval movements here, this so-called feeding stage was designated as static. In contrast, the larvae typically left the media and moved around the well surface during late L3 (*Figure 2A*). This so-called wandering stage was designated as dynamic, as there were consecutive high-activity wandering larval signals (*Figure 2B*), and its duration was 12–24 hr. Thereafter, L3 larval activity gradually decreased, and pupariation followed. During the pupal stage (~100 hr), the activity was not detected in a static phase. The second signal wave activity was detected just after dynamic eclosion.

Sapphire independently and automatically detects maximum peaks at the pupariation and eclosion transition points using machine learning based on an FCN to detect larval and adult animals separately (*Figures 1C* and *2B*). To verify Sapphire's accuracy, we compared pupariation and eclosion timings analyzed by Sapphire and those manually detected (visual handling) from the same images. The Sapphire and visual handling values were nearly identical (*Figure 2C,D*), 73.8% on pupariation and 84.2% on eclosion were only a slight difference within ± 10 frames between them (*Figure 2—figure supplement 2*). Film surface contamination after long-term rearing in small wells might have accounted for the observed decrease in Sapphire detection accuracy. Therefore, to investigate whether Sapphire's detection accuracy improves in a more transparent state of the lid, flies were transferred to the new microplate just after the time that all flies pupated (P14 stage) and scanned time-lapse scanning of the plates. The experiment resulted in 94.7% of values from Sapphire showed nearly identical (within ± 10 frames) to visual handling values, indicating that Sapphire's accuracy was greatly improved relative to the experimental setup in which the same microplate was used both for pupariation and eclosion detections (*Figure 2—figure supplement 3A,B*).

Arbitral image sequences such as raw images and the segmentation images could be analyzed in Sapphire (see Appendix 8). Sapphire calculates the CF signal obtained from the subtraction of consecutive segmentation images as fully automated method (CF method). Sapphire determined the timing of life-event by applying just thresholding on the arbitral time series data specified by user (TH method). At first, threshold is automatically calculated as an average of maximum and minimum value of the signal (auto-TH) and the threshold is modifiable by hands depending on user's demand (manual-TH). Accuracy of CF method and auto- and manual- TH method on raw image subtraction was quantified in (Appendix 8). We compared CF and TH using the same time-lapse image set (*Figure 2—figure supplement 3*). The output of CF was superior to that of TH, as the former is relatively more sensitive to phase shifts. Thus, Sapphire is functional and highly invaluable in automatic analyses.

The relationship between pupariation and eclosion determined by DIAMonDS clearly revealed sexual dimorphism during development (*Figure 2E*). On average, females eclosed 4 hr earlier than males. A previous study corroborated this observation (*Bainbridge and Bownes, 1981*). The pupariation transition time points did not differ between sexes. Therefore, sexual dimorphism in the eclosion time points reflects the difference in pupal duration between sexes (*Figure 2E–G*). No

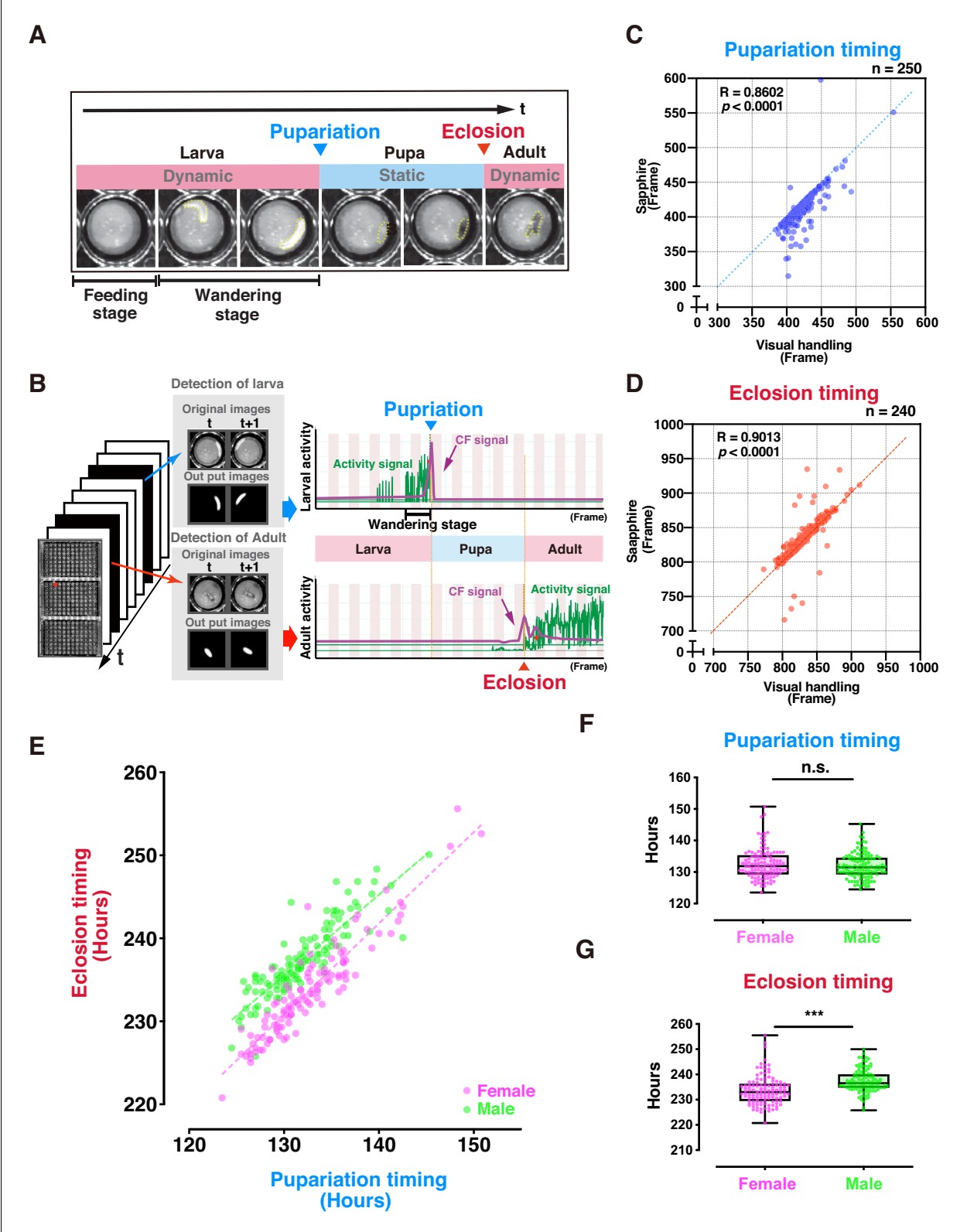

**Figure 2.** Evaluation of DIAMonDS for automatic detection of individual fly pupariation and eclosion. (A) Schematic representation of pupariation and eclosion. Drastic changes in dynamic-to-static and static-to-dynamic states occur at pupariation and eclosion, respectively. At late L3, larval activity increases and transitions from feeding to wandering behavior. Dotted circles indicate animal bodies. (B) Time-lapse imaging was conducted until all flies eclosed into adults. Individual pupariation and eclosion transition points were separately analyzed in Sapphire. Wandering L3 larva showing high

*Figure 2 continued on next page*

*Figure 2 continued*

activity immediately before pupariation. (C,D) Scatterplot analyses comparing data for pupariation (C) and eclosion (D) obtained by Sapphire and visual handling to validate accuracy. (E–G) Scatterplot between pupariation and eclosion timing of individual flies (E) and box plots of pupariation (F) and eclosion (G) timing in males (n = 122; green dots) and females (n = 118; magenta dots). Whiskers indicate minima and maxima (***p<0.001; n.s., no significant difference; unpaired *t*-test).

The online version of this article includes the following source data and figure supplement(s) for figure 2:

**Source data 1.** Data of individual pupariation timing obtained by Sapphire and visual handling (for *Figure 2C* and *Figure 2—figure supplement 2A*).
**Source data 2.** Data of individual eclosion timing obtained by Sapphire and visual handling (for *Figure 2D* and *Figure 2—figure supplement 2B*).
**Source data 3.** Data for scatterplot between pupariation and eclosion timing of individual flies and for box plots of pupariation and eclosion timing in males and females (for *Figure 2E–G*).
**Figure supplement 1.** Effects of experimental conditions.
**Figure supplement 1—source data 1.** Effects of experimental conditions.
**Figure supplement 2.** Residual plot analysis of pupariation and eclosion time points.
**Figure supplement 3.** Validation of CF and TH methods.
**Figure supplement 3—source data 1.** Validation of CF and TH methods.
**Figure supplement 4.** Validation of plate position effect.
**Figure supplement 4—source data 1.** Validation of plate position effect.
**Figure supplement 5.** Effect of plate well size on fly's development.
**Figure supplement 5—source data 1.** Effect of plate well size on fly's development.

significant differences were found between plates 1 and 3 used here, indicating a minimal scan surface positional effect in DIAMonDS during the detection of the pupariation and eclosion time points (*Figure 2—figure supplement 4*). Altogether, these results indicate that DIAMonDS is suitable for automatic measurement of both the pupariation and eclosion time points at high temporal resolution.

To understand the effect of chamber size on fly development, we used three different sizes of microplates (96-well, 48-well, and 24-well) contained normal fly media for measuring pupariation and eclosion. We observed that timings of pupariation and eclosion were slightly but significantly shorten both in the 48-well and 24-well conditions and the pupal duration might have the fewer effects of well-size, suggesting that the chamber size might affect fly's development (*Figure 2—figure supplement 5*).

## DIAMonDS enables autonomic measurement of pupariation and eclosion timing at high temporal resolution for each individual

DIAMonDS showed excellent performance in large-scale pupariation timing analyses (*Figure 3—figure supplement 1*). To evaluate its performance, we explored whether two distinct genetic and environmental conditions affect larval development. First, we used larvae with delayed pupariation at 29°C (genotype: *R29H01-GAL4 > UAS TeTxLC*), in which *tetanus toxin light chain* (*TeTxLC*) impaired serotonergic SE0$_{PG}$ neuron activity (*Shimada-Niwa and Niwa, 2014*). To evaluate the influences of initial larval age on DIAMonDS measurement accuracy, we used L2, early L3, and late L3 larvae in the microplate assays. DIAMonDS successfully detected pupariation delays in larvae of all ages (*Figure 3A*). Thus, DIAMonDS performance remains highly stable over a wide range of conditions.

We also used DIAMonDS to clarify the relationship between sugar concentration and development in the *w*$^{1118}$ fruit fly strain (*Figure 3B–F*). A previous report stated that *Drosophila* larvae that were administered a high-sugar diet presented a type 2 diabetes-like phenotype and developmental delay (*Musselman et al., 2011*). L1 larvae were placed in wells containing normal to high-sugar concentrations, scanned by time-lapse imaging, and the pupariation and eclosion time points were detected by Sapphire. Both pupariation and eclosion were gradually delayed in a sugar concentration-dependent manner. Therefore, excess sugar adversely affects larval growth (*Figure 3B–E*). When we compared the pupariation and eclosion time points among individual larvae, the pupal durations were nearly constant regardless of sugar concentration. Thus, the sugar concentration determines the time spent as a larva until pupariation, but it does not influence the time spent as a pupa (*Figure 3F*). Interestingly, Northrop reported that prolongation of the pre-imago stage of *D. melanogaster* by yeast supplementation had no impact on pupal stage duration (*Northrop, 1917a*; *Northrop, 1917b*). The process by which pupal duration is determined may be independent of larval

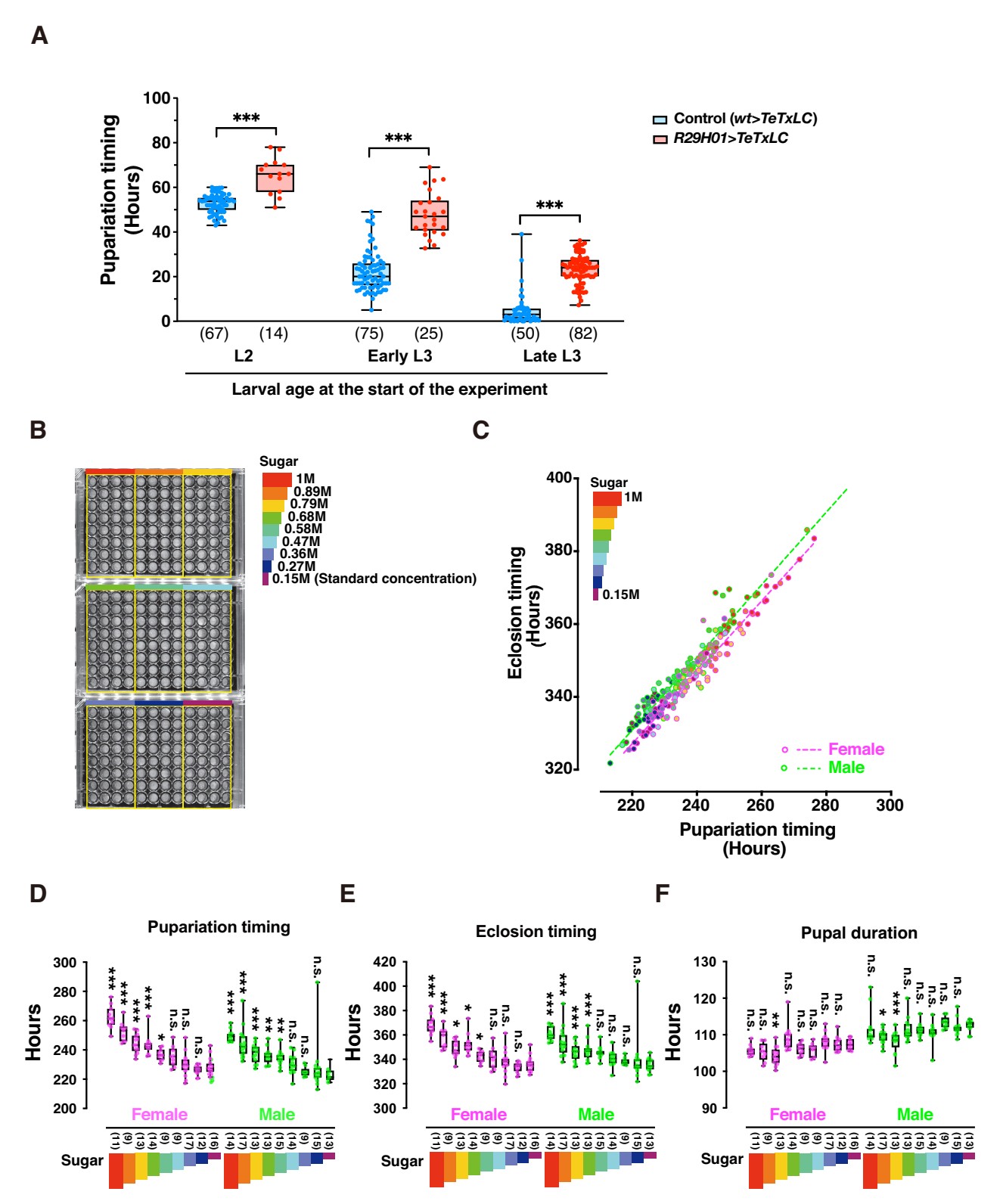

**Figure 3.** DIAMonDS is suitable for phenotypic analyses during larval development. (A) Box plot analysis of pupariation timing in flies with impaired ecdysteroid biosynthesis at 29°C (genotype: *R29H01 > TeTxLC; R29H01>+* as a control). Larval age (L2, early L3, and late L3) at start of measurement had negligible impact on DIAMonDS analysis accuracy. Y-axis indicates pupariation timing from start of experiment. Number of flies analyzed indicated in parentheses on the graph. Whiskers indicate minima and maxima (*p<0.05; **p<0.01; ***p<0.001; n.s., no significant difference; multiple *t*-test). (B) *Figure 3 continued on next page*

*Figure 3 continued*

Three 96-well microplates were subdivided into nine regions according to sucrose concentration (0.15–1 M) in media used for DIAMonDS. (**C**) Scatterplot between pupariation and eclosion timing in males and females. (**D–F**) Box plots of pupariation timing (**D**), eclosion timing (**E**), and pupal duration (**F**). Whiskers indicate minima and maxima (*p<0.05; **p<0.01; ***p<0.001; n.s., no significant difference vs. standard diet group; one-way ANOVA followed by Dunnett's multiple comparison test). Number of flies analyzed indicated in parentheses on the graph.

The online version of this article includes the following source data and figure supplement(s) for figure 3:

**Source data 1.** Box plot analysis of pupariation timing in flies with impaired ecdysteroid biosynthesis at 29˚C (for *Figure 3A*).
**Source data 2.** Effect of sucrose concentration in media on pupariation and eclosion (for *Figure 3C–F*).
**Figure supplement 1.** Determination of large-scale pupariation timing using DIAMonDS.
**Figure supplement 1—source data 1.** Determination of large-scale pupariation timing using DIAMonDS.

dietary intake. Overall, DIAMonDS is a powerful toolkit for detecting the pupariation and eclosion time points and discloses the effects of several endogenous and exogenous factors on individual fly development.

## Detection of individual adult death events by DIAMonDS

Adult death time points can be measured at high temporal resolution with DIAMonDS. This tool efficiently detects sudden shifts from a dynamic to a static phase. Depending on the objective, experiments are performed over a broad range of time scales extending to nearly 3 months. Small- and large-scale impact assessment targets may include mutant phenotype, environmental stress, and chemicals and drugs (*Afschar et al., 2016*; *Harshman et al., 1999*; *Lin et al., 1998*; *Parkes et al., 1998*; *Piper and Partridge, 2016*; *Tsuda et al., 2010*; *Ziehm et al., 2015*). DIAMonDS may be implemented using 96- and 384-well microplates to accommodate various parameters in death timing detection. The 96-well type is used in long-term detection, whereas the 384-well plate is better suited for short-term (≤10 d) detection and high-throughput assays.

We collected time-lapse scanning images at 1 min intervals for adult flies (4 day post-eclosion) maintained under starvation conditions (70 µL 1% agar/water (w/v)/well) in order to optimize DIA-MonDS for 384-well microplates (*Figure 4A*). After the image data series consisted of dying individual flies, death time points were determined manually (visual manipulation) and by Sapphire. Both procedures generated nearly identical survival curves and were highly correlated (R = 0.9913), indicating the reliability of Sapphire (*Figure 4A–D* and *Figure 4—figure supplement 1*). Next, we examined whether the well positions within a microplate influence the death time points. No significant differences were found between each plate or among the sub-areas within each plate (*Figure 4E* and *Figure 4—figure supplement 2*). We also evaluated adult $w^{1118}$ fly starvation tolerance in 96-well microplates and obtained results similar to those acquired from 384-well microplates (*Figure 4F*). Next, we tried to test whether the DIAMonDS can effectively detect the death time points even in flies showed a very reduced amount of activity. As a hypoactive fly model, we used decapitated females who keep motor skills but hardly moved until their death (*Ejima and Griffith, 2008*). DIAMonDS showed relatively good results that were comparable to the visual results although there was a tendency for the accuracy to reduce slightly in comparison with the case of wild-type flies (*Figure 4—figure supplement 3*). Altogether, these results indicate that DIAMonDS is highly effective at detecting fruit fly death events.

## DIAMonDS performs stress resistance assays with high temporal resolution

To assess the efficacy of DIAMonDS at evaluating fly survival under various stress conditions, we performed a starvation assay on male and female $w^{1118}$ flies (*Figure 5A*). They presented sexual dimorphism in terms of starvation resistance. Moreover, the temporal resolution used here (15 min intervals) was higher than those reported in previous studies (*Figure 5B* and *Figure 5—figure supplement 1A,B*; *Grönke et al., 2010*; *Li et al., 2018*).

We then subjected the flies to various concentrations of dichlorodiphenyltrichloroethane (DDT) to identify possible resistance (*Figure 5C,D*; *Afschar et al., 2016*). To a 384-well microplate, we added media consisting of 5% (w/v) sucrose and 0.5% (w/v) yeast as described in Appendix 1. The 48 $w^{1118}$ males in each well were exposed to 0, 0.005, 0.01, 0.05, or 0.1% DDT (*Figure 5C*). DIAMonDS

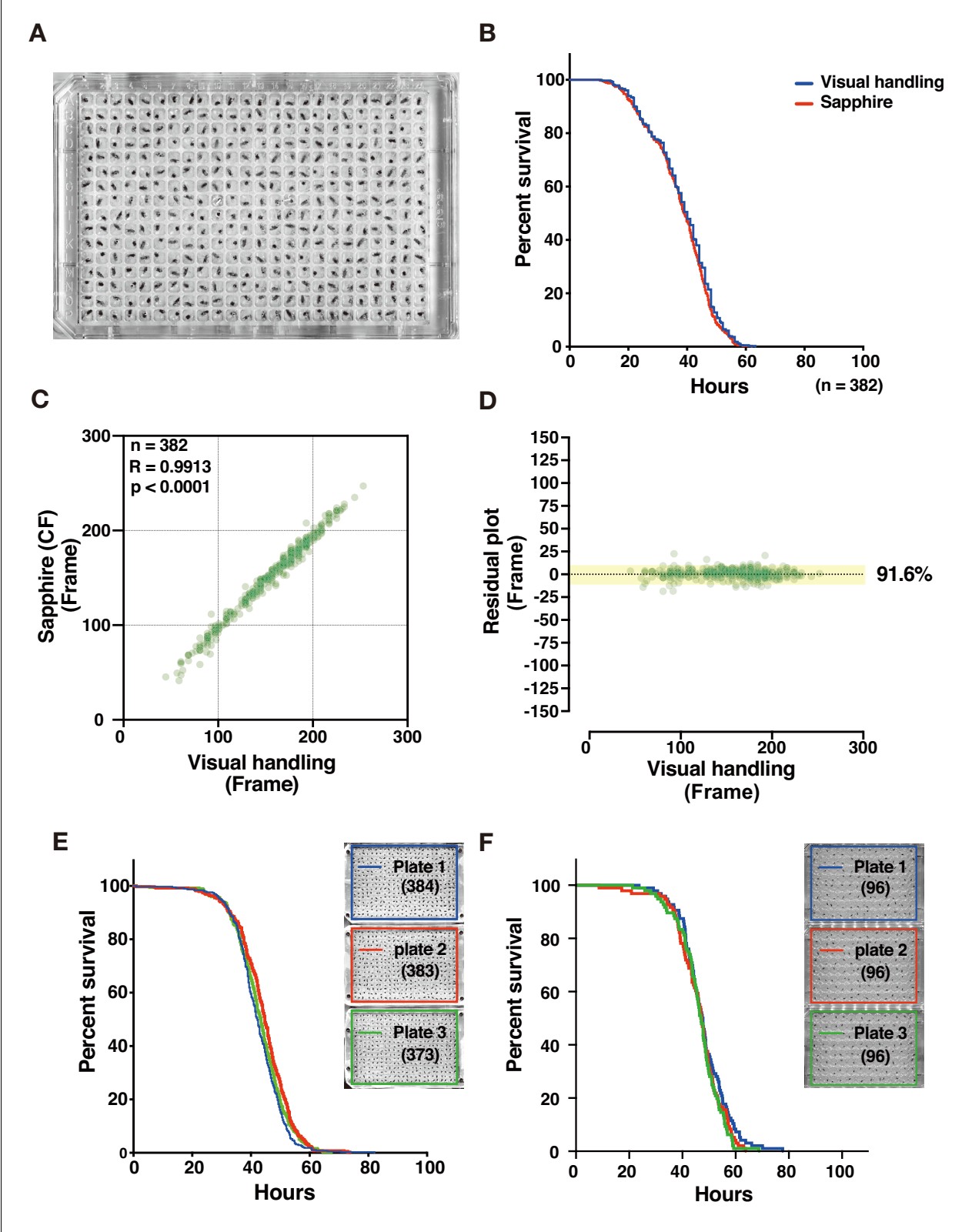

**Figure 4.** Evaluation of DIAMonDS for detection of death of individual adults. (**A**) A 384-well microplate with flies prepared for DIAMonDS. (**B**) Survivorship curves in starvation condition plotted by Sapphire (red line) or visual handling (blue line) using the same data. (**C,D**) Scatterplot and residual plot analysis comparing Sapphire (CF method) and visual handling to validate accuracy (n = 382). (**E,F**) Survivorship curves for starvation

*Figure 4 continued on next page*

*Figure 4 continued*

resistance tests on adult male $w^{1118}$ flies using three 384- (**E**) and three 96-well (**F**) microplates. Number of flies analyzed indicated in parentheses in each plate.

The online version of this article includes the following source data and figure supplement(s) for figure 4:

**Source data 1.** Evaluation of DIAMonDS for detection of death of individual adults.
**Figure supplement 1.** Semi-automatic TH method for Sapphire.
**Figure supplement 1—source data 1.** Semi-automatic TH method for Sapphire.
**Figure supplement 2.** Validation of the effect of sub-area on 384-well microplate.
**Figure supplement 2—source data 1.** Validation of the effect of sub-area on 384-well microplate.
**Figure supplement 3.** Death timing detection of decapitated adult females using DIAMonDS.
**Figure supplement 3—source data 1.** Death timing detection of decapitated adult females using DIAMonDS.

showed distinct variance in DDT resistance even between only slightly differing DDT concentrations (*Figure 5D* and *Figure 5—figure supplement 1C,D*).

We also investigated whether DIAMonDS can detect mutant fly phenotypes in stress resistance assays. Flies with the loss-of-function mutation *methuselah* (*mth*) present extended lifespan and resistance to several stressors including paraquat (*Lin et al., 1998*). To test whether DIAMonDS can discriminate $mth^1$ mutant stress tolerance phenotypes, we backcrossed the $mth^1$ mutant five times with $w^{1118}$ and compared the responses of both types of males to different paraquat concentrations (*Figure 5E,F*). The $mth^1$ mutant presented significantly greater resistance to several paraquat concentrations than $w^{1118}$. The results of this assay demonstrated that paraquat resistance could be detected in the $mth^1$ mutant using lower paraquat concentrations than those tested in an earlier study (20 mM)(*Lin et al., 1998*). Thus, DIAMonDS can readily identify optimal concentrations at high temporal resolution in drug resistance assays.

## Sequential detection of multiple life events (pupariation, eclosion, and lifespan) using DIAMonDS

DIAMonDS successfully measured lifespans for individual $w^{1118}$ flies. For long-term analysis, flies must be transferred to new microplates. We performed sequential pupariation, eclosion, and lifespan measurements for individual $w^{1118}$ as shown in *Figure 6A–C*. The mean lifespans calculated and survival curves plotted for individual males and females were consistent with those of previous reports (*Figure 6C*; *Liu et al., 2009*; *Schriner et al., 2014*; *Suh et al., 2008*; *Trostnikov et al., 2019*).

Several studies described the relationships between developmental time and adult lifespan among various species (*Marchionni et al., 2020*). Certain reports showed that prolonging developmental duration by yeast supplementation did not affect lifespan (*Hunter, 1959*; *Northrop, 1917b*). However, few studies focused on the associations between developmental timing and adult lifespan in *Drosophila*. To establish whether developmental stages and adult lifespans are interdependent in individual flies, we analyzed the following relationships: larval and pupal duration (*Figure 6D,G*), larval duration and adult lifespan (*Figure 6E,H*), and pupal duration and adult lifespan (*Figure 6F,I*). No significant correlations between the data for these pairs of parameters in either sex (*Figure 6D–I*) and those reported in previous studies were found. As the flies had the same genetic background and were exposed to the same diet and environmental conditions throughout all their life stages, it was unlikely that their genotypes influenced the relationship between development time and lifespan, demonstrating that DIAMonDS can detect unique relationships between developmental time and lifespan under different intrinsic and extrinsic conditions.

In this study, we were unsuccessful at using DIAMonDS to run long-term lifespan assays on other wild type fruit fly strains such as *Oregon-R* and *Canton-S*. Lethality accidentally escalated because the rearing conditions deteriorated. For instance, water droplets condensed and coalesced on the plate surfaces. However, these errors might have reflected relative metabolic and/or genetic differences among fly strains. To find common optimum conditions for measuring the lifespan of other laboratory strains, we have tried to observe the viability of three strains ($w^{1118}$, *Oregon-R*, and *Canton-S*) in condition with several microplate well-size (96-well, 48-well, and 24-well) (*Figure 6—figure supplement 1*). The viabilities were measured at the time point at 1, 2, 3, and 4 weeks with once

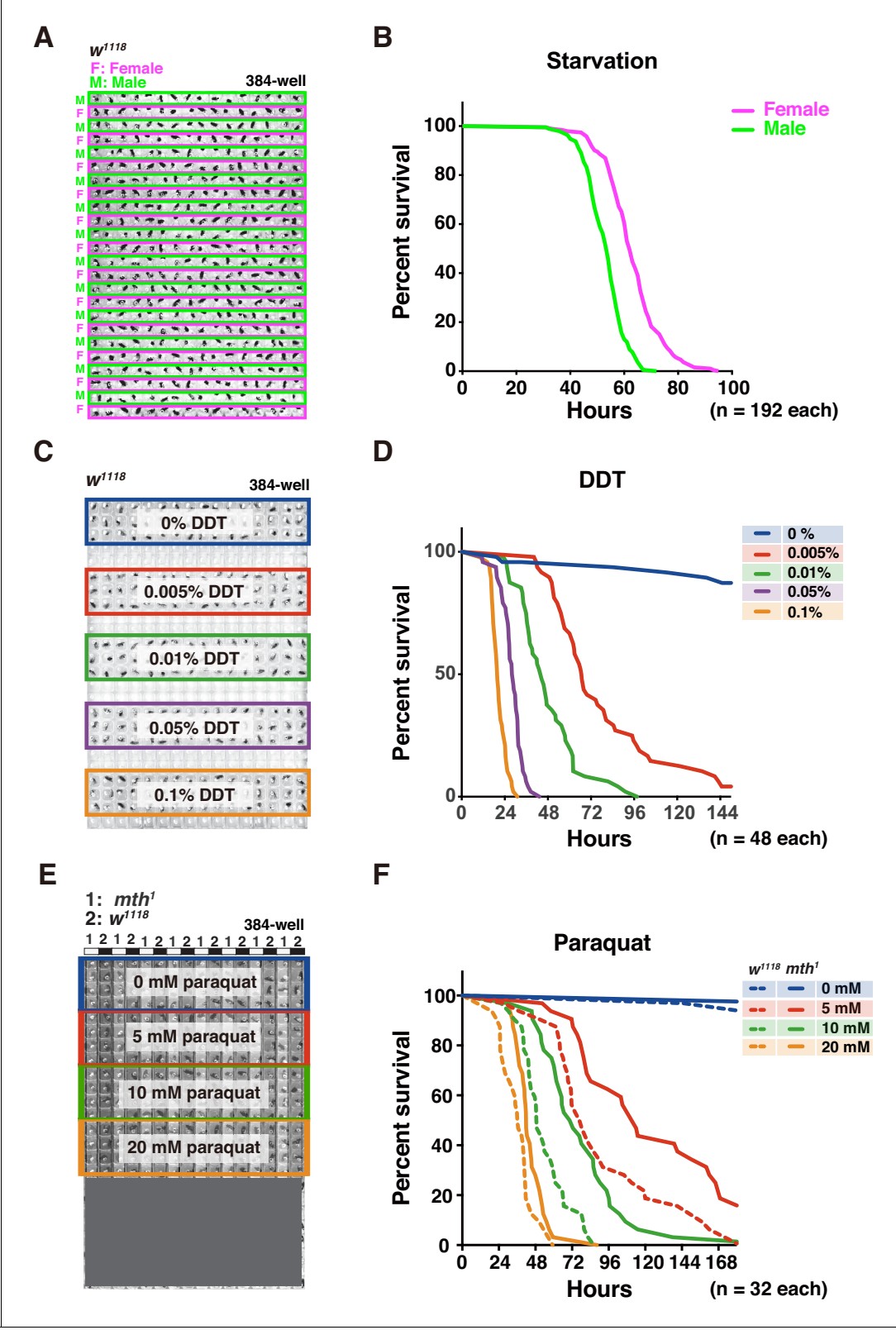

**Figure 5.** Adult survival curve detection by DIAMonDS. (**A,B**) Starvation tolerance test using male and female $w^{1118}$ flies (n = 192 each). Rows of males and females were alternately arranged in 384-well microplate (**A**). Representative male and female survivorship curves (**B**). (**C,D**) DDT resistance test on male $w^{1118}$ flies (n = 48 each). A 384-well microplate with YS media containing DDT concentration series (0–0.1%). Forty-eight male flies were exposed to each concentration and subjected to DIAMonDS (**C**). Survivorship curves show concentration-dependent toxic effects of DDT (**D**). (**E,F**) Paraquat

*Figure 5 continued on next page*

*Figure 5 continued*

resistance test on male *mth¹* and *w¹¹¹⁸* flies (n = 32 each). A 384-well microplate containing media with paraquat (0, 5, 10, and 20 mM). Rows of *mth¹* and *w¹¹¹⁸* flies were alternately arranged in wells (E). Survivorship curves for *mth¹* mutants (solid lines) and *w¹¹¹⁸* flies (dotted lines) substantially differed at all paraquat concentrations (F).

The online version of this article includes the following source data and figure supplement(s) for figure 5:

**Source data 1.** Adult survival curve detection by DIAMonDS (for *Figure 5* and *Figure 5—figure supplement 1*).
**Figure supplement 1.** Comparison between CH and TH methods for detecting death time points in DIAMonDS.

plate replacement a week. We observed that the reduced viability was not rescued by changing to bigger microplate-well size, and somewhat, increasing well-size might have harmful effects on the survival of aged flies (*Figure 6—figure supplement 1A–C*). We next tested the two different plate replacement cycles (once a week, or once 2–3 days) using Oregon-R and Canton-S in 96-well microplate. We observed a shorter plate replacement cycle could effectively improve the viability (*Figure 6—figure supplement 1B,C*). However, because plate replacement is required a lot of work time, it seems to be not suitable for the high-throughput experiments. Further optimization of the experimental conditions will be necessary to effectively use DIAMonDS to conduct lifespan tests on a wide range of fly strains.

## Discussion

In this study, we attempted to develop DIAMonDS as a new tool to analyze automatically and sequentially the transition time point in the growth and developmental multi-stage of individual flies and use the transition time points as a phenotype. We demonstrated that DIAMonDS determines *Drosophila* pupariation, eclosion, and death at high temporal resolution. Further, it can analyze the relationships among stages by sequentially detecting multiple life events in many individuals. Thus, DIAMonDS can help clarify the complex interactions among genetic and environmental factors throughout the *Drosophila* life cycle. DIAMonDS can also eliminate the constraint of long data acquisition and analysis time intervals, operate multiple scanners simultaneously, and facilitate high-throughput analysis. Overall, DIAMonDS substantially ameliorates *Drosophila* research endeavors compared to conventional manual counting methods.

DIAMonDS automatically detects time points from time-lapse images via the novel algorithm Sapphire. Our results indicated that relative to manual detection, DIAMonDS correctly detected 74–85% of the puparation and eclosion and ~ 92% of the death events within ± 10 frames. Thus, DIAMonDS is fully functional both for preliminary experiments and large-scale screening. The output of Sapphire can guide the manual determination of exact values. Therefore, DIAMonDS can generate publishable high-quality data. Detection accuracy could be greatly improved by reducing noise in time-lapse image acquisition. Data quality can also be enhanced via the machine learning training step in Sapphire. Optimization of experimental conditions is an important initial prerequisite step for stable, highly reliable data acquisition.

Nevertheless, DIAMonDS has certain limitations. First, it is unsuitable for the analysis of insects with normal circadian rhythms as the flatbed scanners repeatedly emit light for imaging. Second, the confined well space used for insect rearing might adversely affect individual fly health and behavior. Third, this approach might be inappropriate for establishing the effects of volatile or unstable substances on flies. Fourth, the reliability of this system may be reduced as the surfaces of the microplate lids become dirty over time. Fifth, occasionally, an animal cannot be detected in the image by the existence of a blind area of scanning. The system is not greatly affected even if 'off-camera' occurs during an active state of an animal. But the accuracy of event detection might be slightly affected when 'off-camera' occurs just at the timing of the event shift. It could be reduced by increasing the spatial- or temporal-resolution to eliminate the blind-area of each well. These problems could be solved by further improvement in the system. For example, detection of the circadian rhythm might be achieved by image acquisition using infrared rays, and It seems that the use of large-wells might expect to reduce the adverse effects of rearing on small-wells.

To date, we successfully used DIAMonDS to analyze the normal lifespan of the *w¹¹¹⁸* fruit fly strain but not those of the wild-type *Oregon-R* and *Canton-S*. One reason for this constraint is the frequent occurrence of accidental death caused mainly by water droplet coalescence on the microplate wall

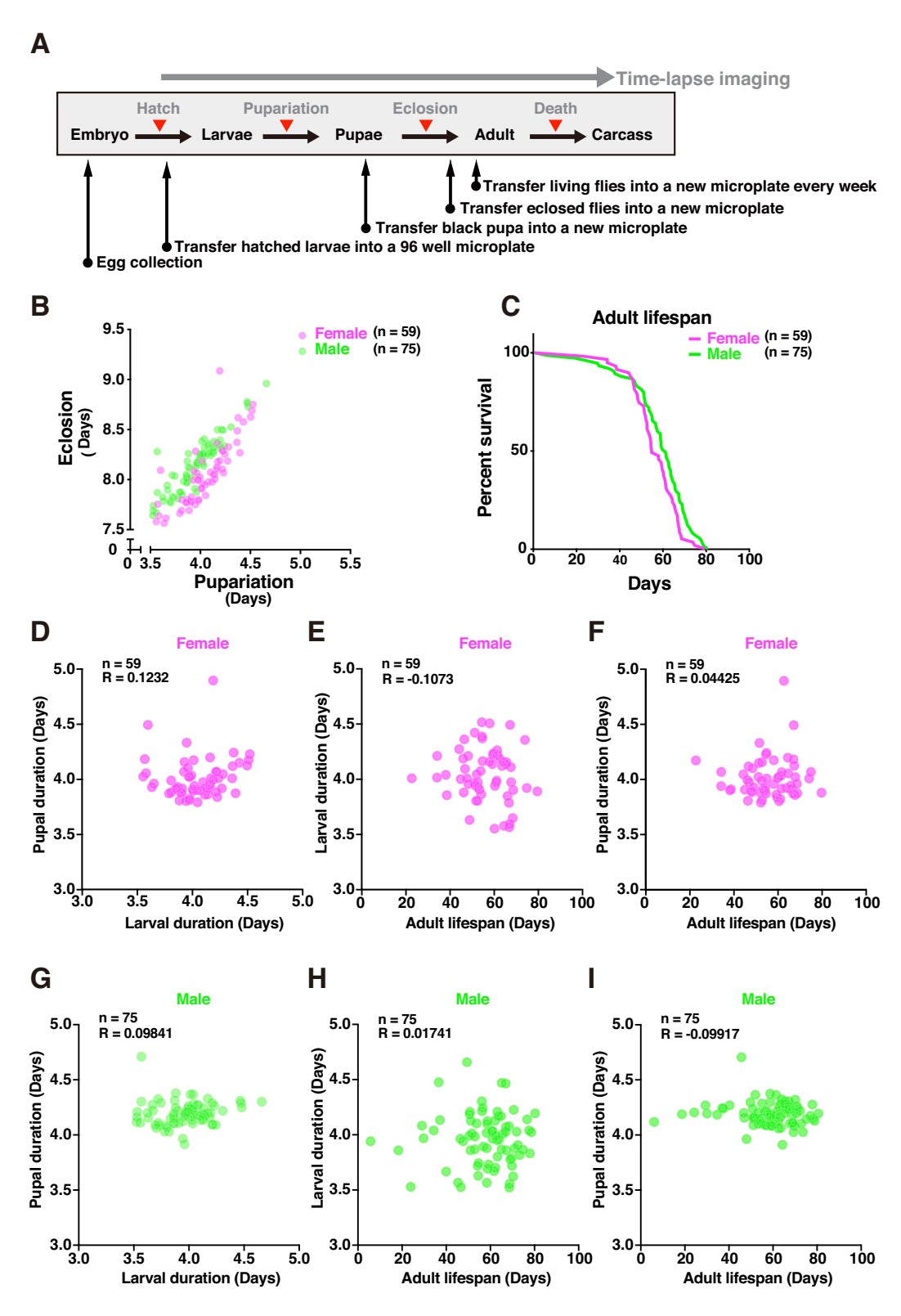

**Figure 6.** Tracing entire life events of individual flies by DIAMonDS. (**A**) Schematic diagram showing detection of pupariation, eclosion, and death transition points for each individual in DIAMonDS. (**B**) Scatterplot between pupariation and eclosion timing for individual female (n = 59) and male (n = 75) flies. (**C**) Survivorship curves for adult female (n = 59) and male (n = 75) flies. (**D–I**) Pearson's correlation scatter plots indicate relationships

*Figure 6 continued on next page*

*Figure 6 continued*

between larval and pupal duration (**D,G**), between larval duration and adult lifespan (**E,H**), and between pupal duration and adult lifespan (**F,I**). Data are separately presented for female (**D–F**, n = 59 each) and male (**G–I**, n = 75 each) flies.

The online version of this article includes the following source data and figure supplement(s) for figure 6:

**Source data 1.** Tracing entire life events of individual flies by DIAMonDS.
**Figure supplement 1.** Effect of plate well size on adult survival.
**Figure supplement 1—source data 1.** Effect of plate well size on adult survival.

surfaces. Differences between strains may be due to differences in genetic background or intestinal environment. The experimental setup will require further optimization to overcome these limitations in lifespan analysis by DIAMonDS.

Here, we reported a novel technique for measuring multiple life events and their transition time points in *Drosophila*. In principle, DIAMonDS can also determine life cycle phase shifts in other small animals, and we intend to expand its applicability in medical, agricultural, and ongoing biological research.

# Materials and methods

## Key resources table

| Reagent type (species) or resource | Designation | Source or reference | Identifiers | Additional information |
|---|---|---|---|---|
| Gene (*D. melanogaster*) | *mth¹* | Flybase | FBti0012557 | |
| Strain, strain background (*D. melanogaster*) | *w¹¹¹⁸* | Kept in lab stock | | |
| Strain, strain background (*D. melanogaster*) | *Oregon-R* | Kept in lab stock | | |
| Strain, strain background (*D. melanogaster*) | *Canton-S* | Gift from Dr. Uemura | | |
| Genetic reagent (*D. melanogaster*) | *R29H01-GAL4* | Flybase | FBti0191124 | |
| Genetic reagent (*D. melanogaster*) | *UAS-TeTxLC* | Flybase | FBtp0001264 | |
| Software, algorithm | Sapphire | | | https://github.com/kanglab/Sapphire/tree/master |
| Software, algorithm | VueScan | | | https://www.hamrick.com |
| Software, algorithm | Prism 8 | | | https://www.graphpad.com/scientific-software/prism/ |

## *D. melanogaster* stocks and rearing conditions

The wild-type strain used here was mainly $w^{1118}$, *Oregon-R,* and *Canton-S*. The $mth^1$ (BDSC #27896) mutant males were used in the paraquat resistance test after backcrossing six-fold with $w^{1118}$. *R29H01-GAL4* (BDSC #47343) and *UAS-TeTxLC* (BDSC #28838) were obtained from the Bloomington *Drosophila* Stock Center, Bloomington, IN, USA. All fly strains were maintained on standard medium at 25°C (Appendix 1).

Most experiments were conducted in a plant growth chamber (LPH-410NS; NK System Co. Ltd., Nagoya, Japan) maintained at 25°C and 60% RH. For the experiment using *R29H01-GAL4 > UAS TeTxLC,* the larvae were reared at 29°C to increase GAL4 activity.

## DIAMonDS hardware and layout

Detailed DIAMonDS hardware and layout are described in Appendix 6.

### Microplate preparation for DIAMonDS

Microplates with 96 or 384 wells were filled with standard fruit fly culture media (Appendix 1). A handmade acrylic lid was used for the 384-well microplate (Appendix 2). Titer stick film (Watson, Tokyo, Japan) was used for the 96-well microplate (Appendix 3). To determine the pupariation and eclosion timings, the film seal on the 96-well microplate was perforated with air holes (0.35 mm diameter; eight holes/well) over each well. For the stress resistance and lifespan tests, the 96-well microplate was sealed with film, which was then cut to form a cross shape over each well. Individual adult flies were transferred to each well without anesthesia. For the 384-well microplate, the flies were anesthetized with triethylamine and placed individually into each well (Appendix 4). The acrylic lid was then securely affixed to the microplate with screws and masking tape (Appendix 2).

### Time-lapse image acquisition

Three microplates were secured with glass slides to the flatbed scanner surface as described in Appendix 5. The scanner was then connected to a personal computer. 'VueScan' scanner software in the PC was launched, and time-lapse images were acquired at 1–15 min intervals (Appendix 6).

### Starvation and drug resistance test

For the drug resistance assay, the wells of a microplate were filled with yeast-sucrose medium containing the appropriate chemicals as shown in Appendix 1. For the starvation test, the wells of a microplate were filled with 1% (w/v) agar (Appendix 1). Virgin adult flies were collected under $CO_2$ anesthesia, stored in a vial with normal food for 3–5 days, placed individually into each microplate well, and subjected to time-lapse image acquisition.

### Pupariation and eclosion time point detection for the same individuals

To measure pupariation and eclosion timing, newly hatched L1 larvae (24–25 hr AEL) were used as the lethality was too high when embryos were used (*Figure 2—figure supplement 1A*). The large media volume in each well (150 µL) also increased lethality because the embryos suffocated when the media became very viscous. Therefore, we used 100 µL media for this approach (*Figure 2—figure supplement 1B*). L2 or L3 larvae were also used in the DIAMonDS experiments. After the larvae were loaded into the microplate wells, the microplate was covered with Titer stick film (Watson, Tokyo, Japan), inverted on a predetermined area of the flatbed scanner surface, and fastened with tape. The scanner was then inverted in the incubator to prevent liquefied culture medium from falling onto the film surface. Time-lapse scanning was then run in VueScan in the PC. Scanning was terminated when all individuals eclosed to adults. The time-lapse images were analyzed in Sapphire (*Figure 1B,C*).

### Pupariation, eclosion, and lifespan detection in identical individuals in DIAMonDS

Microplates were prepared with synchronized L1 larva (one/well), and images were acquired as previously described. Developmental stages for individuals were analyzed according to the images saved to the PC. After all individuals reached the P14 stage pupa, the pupae were transferred to the same well positions in a new microplate. The wells were sealed with new film, and scanning was resumed. After all flies completely eclosed, the individual adults were transferred to the same well positions in a new microplate, and scanning was restarted to measure individual lifespans. Adult flies were transferred weekly to new microplates until all flies died. The acquired images were then analyzed in Sapphire. Finally, we introduced the data obtained by Sapphire into the DIAMonDS analysis templates (*Supplementary file 1*) and calculated the time of each life-event (Appendix 7).

### Life event detection algorithm and software: Sapphire

The present system included an automated, high-accuracy life event detection algorithm and image and signal processing (*Figure 1C*). The software could be obtained from (https://github.com/kanglab/Sapphire/tree/master; *Seong et al., 2020*; copy archived at https://github.com/elifesciences-publications/Sapphire). The contents in the address also describe the installation, license, and system requirements such as directory tree and external dependencies. Quantitative algorithm summary and robustness in various parameter spaces are described in Appendix 8.

## Image processing

The present system performed semantic segmentation via a deep neural network whose architecture is described in *Figure 1—figure supplement 2*. Sufficient annotation data increases inference accuracy in deep learning. For network training, handmade supervised data were prepared and applied to image data augmentation. For adult body segmentation in death detection, the annotation data for 178 adults were introduced and magnified up to 71,200 by general image data augmentation (vertical and horizontal flips, rotation, and luminance modulation). For adult body segmentation in pupariation and eclosion detection, annotation data for 300 adults and 5760 larvae (as distractors) were prepared and magnified by $\leq$ 60,000 and$\leq$57,600, respectively. For larval body segmentation in pupariation and eclosion detection, annotation data for 1839 larvae and 183,900 augmented data were introduced to train the network. The present system detected rough outlines of the animal bodies (Appendix 8).

## Signal processing

In the present system, subtractions between consecutive segmentation images were converted to signals in CF, which is a change point detection algorithm (*Takeuchi and Yamanishi, 2006*).

The CF signal was obtained as follows:

1. Considering the following AR model for time series data $x_i$:

$$x_t = A_{t-1}x_{t-1} + A_{t-2}x_{t-2} + \cdots \tag{1}$$

2. Inference of parameters $\theta = (A_1, \cdots, A_k, \mu, \sigma)$ by maximizing $I$:

$$I = \sum_{i=1}^{t}(1-r)^{t-1}\log P(x_i|x_{i-1}, \theta) \tag{2}$$

3. Calculation of scores:

$$Score(x_t) = -\log P_{t-1}(x_i|x_{i-1}) \tag{3}$$

4. Time averaging of scores:

$$y_t = \frac{1}{T}\sum_{i=t-T+1}^{t} Score(x_i) \tag{4}$$

5. Recalculating scores for $y_t$:

For death detection, the CF signal was calculated from the adult body segmentation and the death timing and determined as a maximum CF signal point capturing dynamic-to-static changes (Appendix 8). On the other hand, two CF signals were obtained from the adult and larval body segmentations in pupariation and eclosion detection. Pupariation is defined as the maximum dynamic-to-static transition point for larvae. Eclosion is defined as the maximum static-to-dynamic transition point for adults (Appendix 8). After time-lapse scanning, we can easily discriminate 'out-of-event', such as larval, and, pupal dead individuals. Sapphire can exclude 'out-of-event' wells by making the 'black lists' file (Appendix 7).

## Statistics

Data were analyzed, and graphs were plotted in GraphPad Prism v. 8 (GraphPad Software, San Diego, CA, USA). A student's unpaired, two-tailed *t*-test was performed to compare differences between groups in each experiment and Dunnett's method of one-way ANOVA was used for multiple comparison test (***$p<0.001$; **$p<0.01$; *$p<0.05$; n.s., no significant).

## Acknowledgements

We are deeply grateful to Tadashi Uemura (Kyoto University) for comments, discussion and helpful supports. We thank Yuko Iijima, Eisuke Imura and Hsin Kuang Lin for their technical assistance. We are also grateful to Seino Masaki, Maki Otori, Mayu Kudo, Masaya Hata and Hiroki Watanabe for

their technical support in image processing. I received generous support from Yoichi Shinkai (RIKEN CPR) and Shunsuke Ishii (RIKEN CPR).

## Additional information

### Funding

| Funder | Grant reference number | Author |
| --- | --- | --- |
| Japan Agency for Medical Research and Development | JP18gm1110001 | Ki-Hyeon Seong<br>Ryusuke Niwa<br>Siu Kang |
| Japan Science and Technology Agency | JPMJPR12M5 | Ki-Hyeon Seong |
| Japan Society for the Promotion of Science | 19K06497 | Ki-Hyeon Seong |

The funders had no role in study design, data collection and interpretation, or the decision to submit the work for publication.

### Author contributions

Ki-Hyeon Seong, Co-last author (jointly supervised this work), Conceptualization, Data curation, Software, Formal analysis, Supervision, Funding acquisition, Validation, Investigation, Visualization, Methodology, Writing - original draft, Project administration, Writing - review and editing; Taishi Matsumura, Data curation, Software, Formal analysis, Validation, Investigation, Visualization, Methodology, Writing - original draft; Yuko Shimada-Niwa, Formal analysis, Validation, Investigation, Writing - review and editing; Ryusuke Niwa, Funding acquisition, Investigation, Writing - review and editing; Siu Kang, Co-last author (jointly supervised this work), Data curation, Software, Formal analysis, Supervision, Funding acquisition, Validation, Investigation, Visualization, Methodology, Writing - original draft, Project administration, Writing - review and editing

### Author ORCIDs

Ki-Hyeon Seong (iD) https://orcid.org/0000-0002-0606-8896
Yuko Shimada-Niwa (iD) http://orcid.org/0000-0001-5757-4329
Ryusuke Niwa (iD) http://orcid.org/0000-0002-1716-455X

### Ethics

Animal experimentation: This study was performed in strict accordance with the recommendations in the Guide for the Care and Use of Laboratory Animals of the RIKEN Institute.

### Decision letter and Author response

Decision letter https://doi.org/10.7554/eLife.58630.sa1
Author response https://doi.org/10.7554/eLife.58630.sa2

## Additional files

### Supplementary files

- Supplementary file 1. DIAMonDS analysis templates.
- Transparent reporting form

### Data availability

To identify the phase change time points for each fly event, we have used the Sappire, our newly developed algorithm and web application. We have deposited the code of Sapphire in the GitHub (https://github.com/kanglab/Sapphire/tree/master copy archived at https://github.com/elifesciences-publications/Sapphire). The all statistical analyses have used Prism 8 software.

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

## Appendix 1

### Fly Media for DIAMonDs

### 1.1. Materials and reagents

Agar powder: WAKO 010–15815 (500 g)
Agarose L (Low melting type): WAKO 317–02282 (25 g)
D(+)-Glucose: WAKO 045–31167 (10 Kg)
Sucrose: WAKO 193–00025 (500 g)
Dry yeast: Beer Yeast Korea Inc Dry Yeast G2 (20 Kg)
Cornmeal: SUNNY MAIZE CO.,LTD. Corn grits No. 4M (25 Kg)
96-well microplate: TrueLine TR5003 96well culture plate
384-well microplate: Greiner Bio-One 781186 Clear Polystyrene 384 well Microplate
Titer stick film: WATOSON 547-KTS-HC
Fly rearing plastic vial: MKC-30 [small] (HIGH TECH LLC, Chiba, Japan)
Soft sponge plugs: COW 30t × 27Ø; (HIGH TECH LLC, Chiba, Japan)

### 1.2. Standard fly medium

- Ingredients
    Agar powder 21 grams
    D(+)-Glucose 300 grams
    Dry yeast 150 grams
    Corn Grits 201 grams
    Mili-Q water 3.5 liters
    Propionic acid 10 milliliters
- Instructions
    1. Measure each material and mix all.
    2. Stir well with heating by a table-top gas cooker until materials fully dissolved.
    3. Autoclave the medium.
    4. Cool down the medium up to 80℃
    5. Add 10 ml propionic acid and mix well. (For fly rearing vial)
    6. Dispense the medium into each vial (5 ml/vial).
    7. Cool before plugging. (For preparation of DIAMonDS microplate with standard medium)
    8. Dispense the medium into each well of 96-well microplate (add 170 µl for lifespan test or 100 µl for detection of pupariation and eclosion).

### 1.3. Yeast-sucrose (YS) medium for DIAMonDS drug resistance test

We have used YS medium for drug resistance test of DIAMonDS. We found that addition of 0.5% of dry yeast in 5% sucrose media showed highest survival and flies kept a high survival rate at least 10 days in the well of 384-well microplate with the food condition (*Appendix 1—figure 1*), it seems to enough to the short-term test such as stress resistance and chemical toxicity test using 384-well microplate.

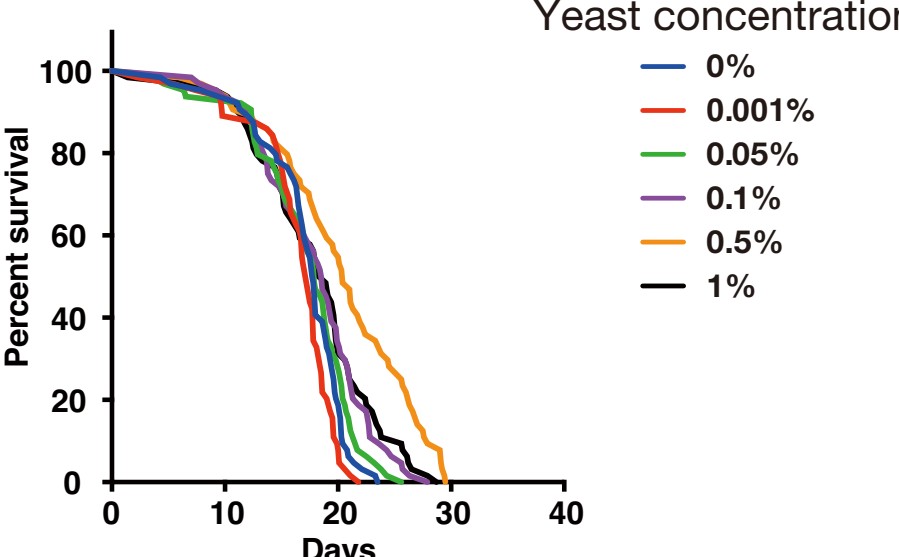

**Appendix 1—figure 1.** Effect of yeast concentration in fly medium. Survivorship curves for $w^{1118}$ males under several yeast concentrations by using a 384-well microplate (n = 24 each).

- Ingredients
    - Agarose L 7.5 grams
    - Sucrose 25 grams
    - Dry yeast 2.5 grams
    - Mili-Q water (up to 500 milliliters)
    - Propionic acid 1.5 milliliters
- Instructions
    1. Measure each material and mix all.
    2. Autoclave the medium.
    3. Cool down and keep the medium at 50℃ by water bath.
    4. Add 1.5 ml propionic acid and mix well.
    5. Add appropriate volume of chemical drug and mix well.
    6. Dispense the medium into each well (70 µl for 384-well and 170 µl for 96-well microplate).
    7. Cool down and leave the plate until the medium surface is dry.

## 1.4. Agar plate for starvation

- Ingredients
    - Agar powder five grams
    - Mili-Q water (up to 500 milliliters)
- Instructions
    1. Measure each material and mix all.
    2. Autoclave the medium.
    3. Cool down the medium up to 80℃
    4. Dispense the medium into each well (70 µl for 384-well and 170 µl for 96-well microplate).
    5. Cool down and leave the plate until the medium surface is dry

## Appendix 2

### Lid for 384-well microplate

384-well microplate: Greiner Bio-One 781186 Clear Polystyrene 384 well Microplate

We have made a lid for 384-well microplate from an acrylic sheet (2 mm thick) as shown in *Appendix 2—figure 1a*. Holes (diameter = 0.7 mm) were drilled in each well position with an electric drill. The four corners were fixed by screws, and all edges were sealed by masking tape (*Appendix 2—figure 1b*).

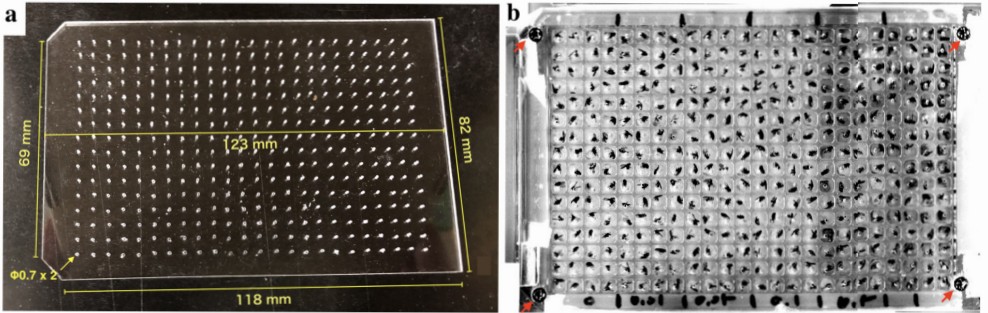

**Appendix 2—figure 1.** Lid design for 384-well microplate. (a) A handmade acrylic lid for 384-well microplate. (b) Usage example of the lid. Red arrows indicate positions of the screw.

## Appendix 3

### Lid for 96-well microplate

96-well microplate: TrueLine TR5003 96well culture plate
  Titer stick HC film: WATOSON 547-KTS-HC

### 3–1. To measure the developmental time points of pupariation and eclosion

The titer stick film is covered on the 96-well microplate. Then, the plate is sealed on the upper surface by microplate sealing titer stick film (547-KTS-HC, WATOSON bio lab), and make eight air holes (0.35 mm) on the film surface of each well by insect pin (No. 00).

### 3–2. Stress resistance test and lifespan test

To enable transfer flies into the well in no anesthetized treatment, the cross-shaped cutting on film surfaces on each well is conducted by a sharp cutter after sealing the microplate by the film. Then, individual adults were transfers to each of the well by using a fly aspirator without the use of anesthetized treatment.

## Appendix 4

### Anesthetized treatment

The use of anesthetics can reduce the labor dramatically for inserting individual flies in each well, especially in the case of a 384-well microplate. We tested two anesthetics, carbon dioxide and triethylamine (TEA, also called as FlyNap) (*Fuyama, 1977*). Short exposure of CO2 (10, 30, and 60 s) or TEA (10, 20, 30 s) did not show a significant difference in comparison with no treated flies (*Appendix 4—figure 1a and b*). The narcotism by short exposure of TEA vapor continues for 30 min (*Fuyama, 1977*). Therefore, we have been preferentially using TEA for the preparation of a fly plate.

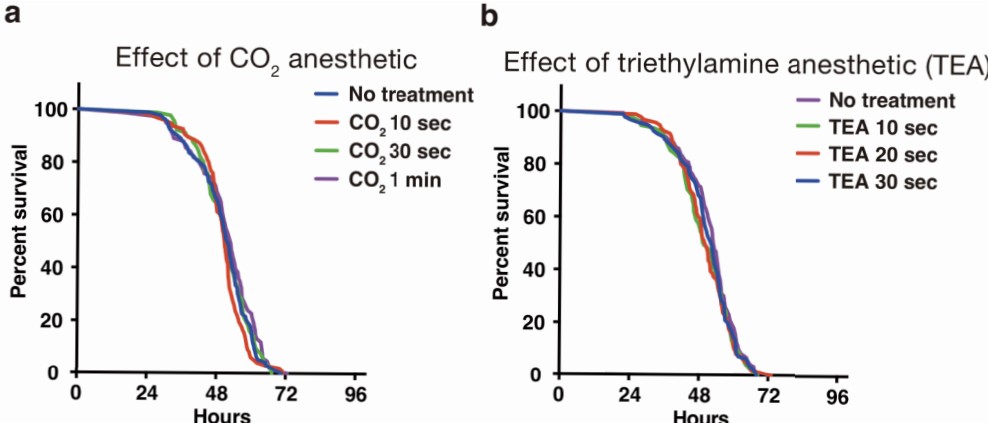

**Appendix 4—figure 1.** Effect of anesthetics on fly viability in starvation test of adult male flies using 384-well microplate. (**a**) Effect of $CO_2$ anesthetic. (**b**) Effect of TEA anesthetic.

## Appendix 5

### Fixing the microplates on the scan surface of the scanner

To put microplates on the fixed position every time, glass slides (MATSUNAMI Glass, S-2215) were arranged and glued on the scan surface as shown in *Appendix 5—figure 1a* (red squares indicate positions of glass slides). The microplates were placed along the guide of slide glasses and sealed by masking tape (*Appendix 5—figure 1b and c*). To reduce the reflection during scanning, black painted sheet was put on the surface of a scanner reflector (*Appendix 5—figure 1d*).

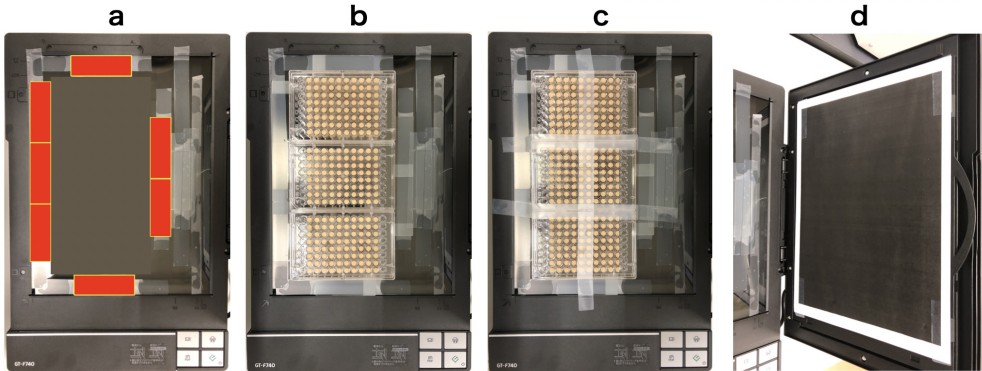

**Appendix 5—figure 1.** Fixing method of microplates on the scanner surface. (**a**) Positions of glass slides (red squares). (**b**) Three microplates fit perfectly in the frame of glass slides. (**c**) Microplates are fixed tightly on the surface by masking tape. (**d**) A black painted sheet on the scanner reflector.

## Appendix 6

### Image acquisition by DIAMonDS

#### 6.1. Equipments and a software

Rearing chamber: LPH-410NS (Nippon Medical and Chemical Instruments Co., Ltd., Japan)
   USB Fan: BIGFAN 120U (TIMELY Co,Ltd., Japan)
   USB Hub: 400-HUB035BK (SANWA SUPPLY Inc, Japan)
   Scanner: EPSON GT-F740 (also called 'Epson Perfection V370 Photo')
   Uninterruptible power supply (UPS): BY120S (OMRON Corporation, Japan)
   PC: Mac mini (Late 2014) (APPLE Inc)
   Scanner software: VueScan (Hamrick Software, USA, https://www.hamrick.com)

#### 6.2. Hardware layout of the DIAMonDS

We have set up the scanner system in the rearing chamber as shown in *Appendix 6—figure 1*. Multiple scanners are connected to a single PC (Mac mini, late 2014, Apple Inc). We have used an Uninterruptible power supply (UPS) (BY120S, OMRON) to reduce the risk of a blackout. To minimize temperature unevenness on the scan surface, multiple small USB-fan (d=12 cm) are installed and operated in the chamber. DIAMonDS could use any CCD flatbed scanner. We have preferentially used EPSON GT-F740.

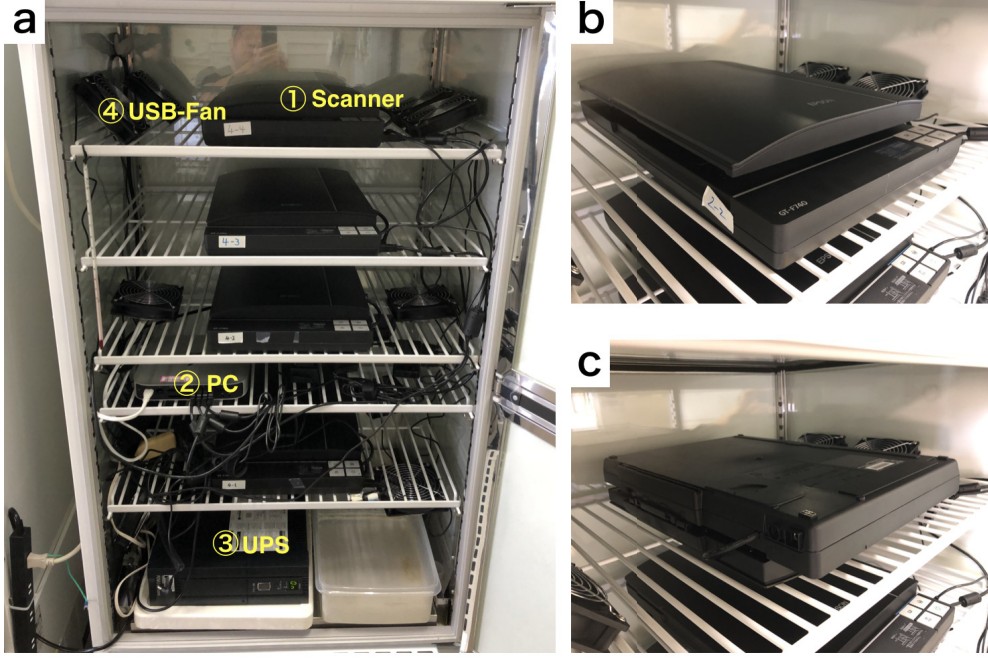

**Appendix 6—figure 1.** Equipment arrangement for DIAMonDS. (**a**) An example of the layout for DIAMonDS. (**b and c**) The direction of installation of the scanner for measurement of death time point. The scanner should be installed in the normal direction or upside-down direction for measurement of death (**b**) or developmental timings (pupariation and eclosion) (**c**) respectively.

For the detection of individual death by DIAMonDS, the scanner is installed in the normal orientation (*Appendix 6—figure 1b*). The medium solidified with agar gradually becomes liquified due to the feeding behavior of larvae. It is a cause of unclearness of the lid surface, and increase data noise. To reduce it, the scanner is oriented upside-down direction for measurement of developmental timings of pupariation and eclosion by DIAMonDS as shown in *Appendix 6—figure 1c*.

## 6.3. Rearing condition

All experiments were conducted in a Plant growth chamber (LPH-410NS) which was kept at 25℃ and at relative 60% humidity. Heterogeneity of temperature on the scan surface might affect data quality. To avoid the problem, several small fans (BIGFAN 120U) were installed into the chamber and the average temperature difference between several positions on scanner were achieved less than 0.5℃ in dark condition (*Figure 1—figure supplement 1*).

## 6.4. Time-lapse image acquisition

To acquire the time-lapse image, we have used a scanner driver, '**VueScan**' (Hamrick Software, USA, https://www.hamrick.com). The VueScan is able to capture images continuously at several interval timing using multiple units of scanners (*Smith et al., 2014*). Using the VueScan, we performed time-lapse imaging as following protocol.

### Time-lapse image acquisition protocol

1. Create new folder for storing images. An example of a folder name is '20190707-Seong-w1118.pupariation-scanner1' ('date-researcher's name-information of experiment-scanner number').
2. Connect the scanner to the PC and turn on the scanner.
3. Open the VueScan.
4. If you already saved the 'options…", you can load the option which recorded previously used whole parameters for acquisition of images.
5. Press the 'preview' button, (press 'cancel' button after completion of the preview), and set a region of interest (ROI) on the previewed image.
6. Set parameters in each tab as shown in *Appendix 6—figure 2* and *Appendix 6—figure 3*. In the tab 'Input': we use 'eight bit Gray', and '150 dpi' both as scan and preview resolutions. Press 'Auto repeat' drop-down box and choose repeat interval as shown in *Appendix 6—figure 2b*. Select the folder to save images. Input the JPEG file name like as '000001+.jpg'. In the tab 'Color': Input the appropriate values in 'Black point (%)", 'white point (%)", 'Curve low', and 'Curve high' to take a high contrast image. In the tab 'Output': confirm each parameter as shown in *Appendix 6—figure 3b*.
7. Press 'File'/"Save options…' and save the parameters as same folder name (e.g.: '20190707-Seong-w1118.pupariation-scanner1').
8. Press 'scan' button.
9. (Option) To perform multiple scanners using one PC, repeat steps 1 to 8 for each scanner sequentially. Before the multiple scanner running, you need to replicate and assign the VueScan application for each scanner. Modify the application name of replicated each VueScan as like 'VueScan-1', 'VueScan-2'... Then, open each VueScan application independently correspond to each scanner respectively.

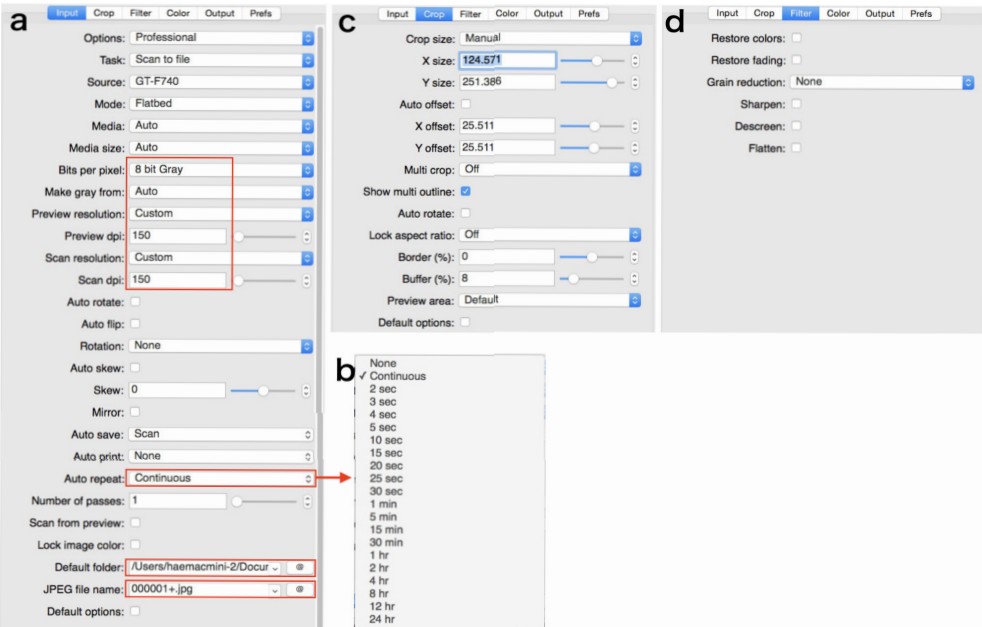

**Appendix 6—figure 2.** Parameters in the tabs. Areas enclosed by the square is especially important. (**a**) 'Input' tab. (**b**) Drop-down box of 'Auto repeat'. (**c**) 'Crop' tab. (**d**) 'Filter' tab.

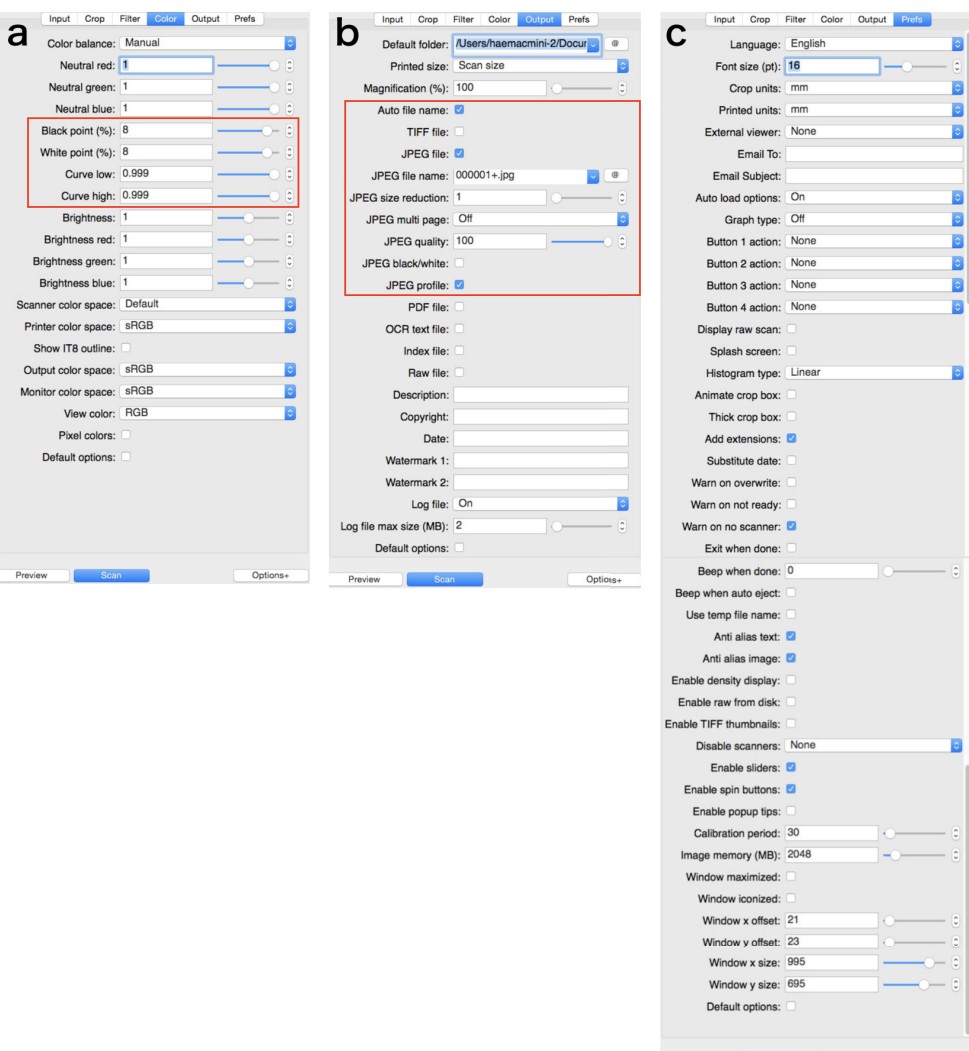

**Appendix 6—figure 3.** Parameters in the tabs. Areas enclosed by the square is especially important. (**a**) 'Color' tab. (**b**) 'Output' tab. (**c**) 'Prefs' tab.

## Appendix 7

### Template for analysis of DIAMonDS

We have prepared templates for data analysis of results from the Sapphire. The Sapphire analyzes the time-lapse images and, as a result, determines the event changing point as the frame number. Next, we should input each values of data ('Blacklists', 'Timestamp', and 'Event timings (auto)") into the appropriate positions of the Excel templates. The procedure is following.

1. Download each data set ('Blacklists', 'Timestamp', and 'Event timings (auto)") from the Sapphire viewer (*Appendix 7—figure 1*).
   'Blacklists' contains the information of omitted wells from the analysis. We can select omitted wells in the main tab on the Sapphire viewer.
   'Timestamp' is a list of date of each image.
   'Event timings (auto)' is frame number list of each well calculated by Sapphire.
2. Copy and paste downloaded data lists into appropriate position on the template (*Appendix 7—figure 2* and *Appendix 7—figure 3*).
3. If necessary, input 'M' or 'F'. You can separate the male ('M') and female ('F') using the sex information of each well (*Appendix 7—figure 3*).

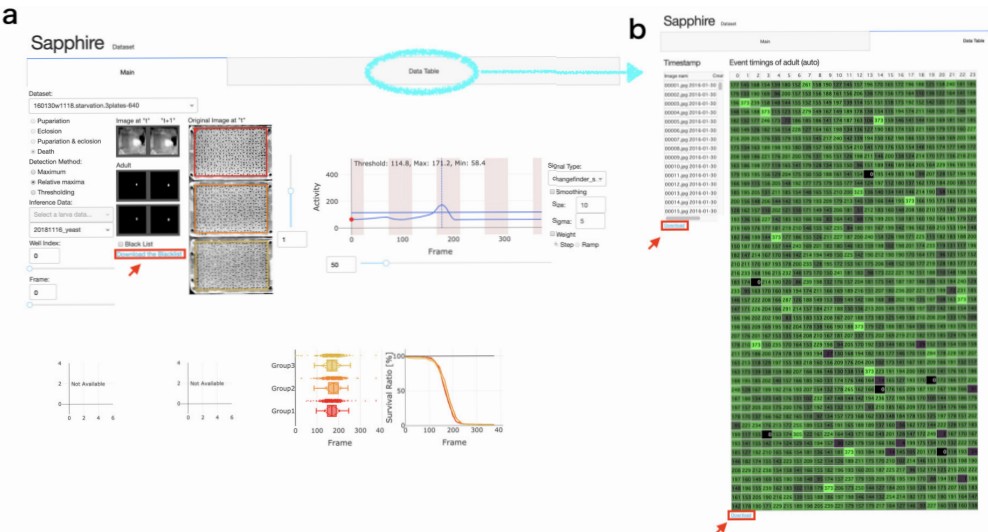

**Appendix 7—figure 1.** Sapphire viewer. (**a**) 'Main' tab. (**b**) 'Data table' tab. Red arrows indicated 'Download' button of each data lists.

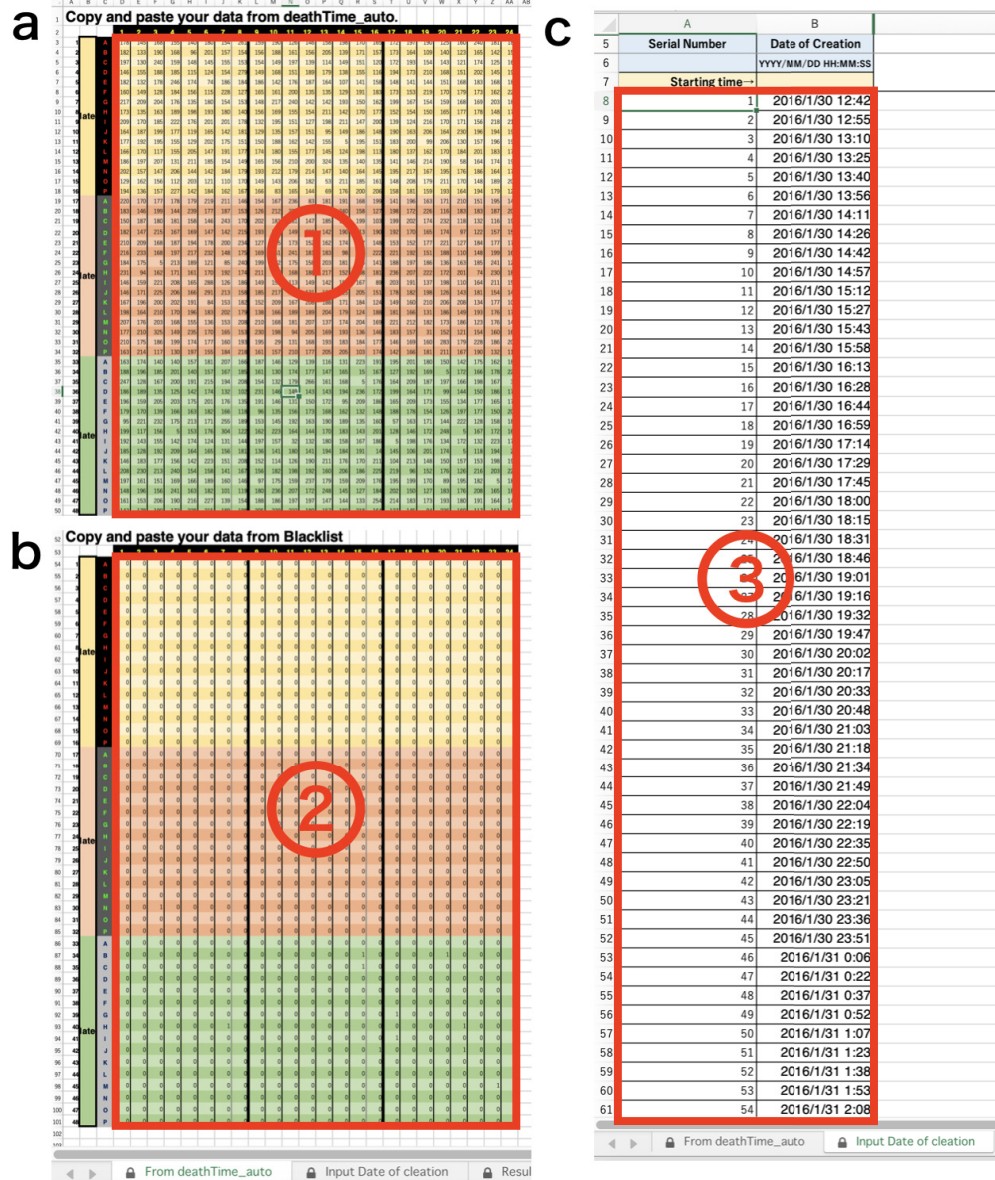

**Appendix 7—figure 2.** DIAMonDS analysis template: tabs for data input. (**a**) Input data list calculated by Sapphire into ①. (**b**) Input 'Blacklist' downloaded from Sapphire into ②. (**c**) Input 'Timestamp' downloaded from Sapphire into ③.

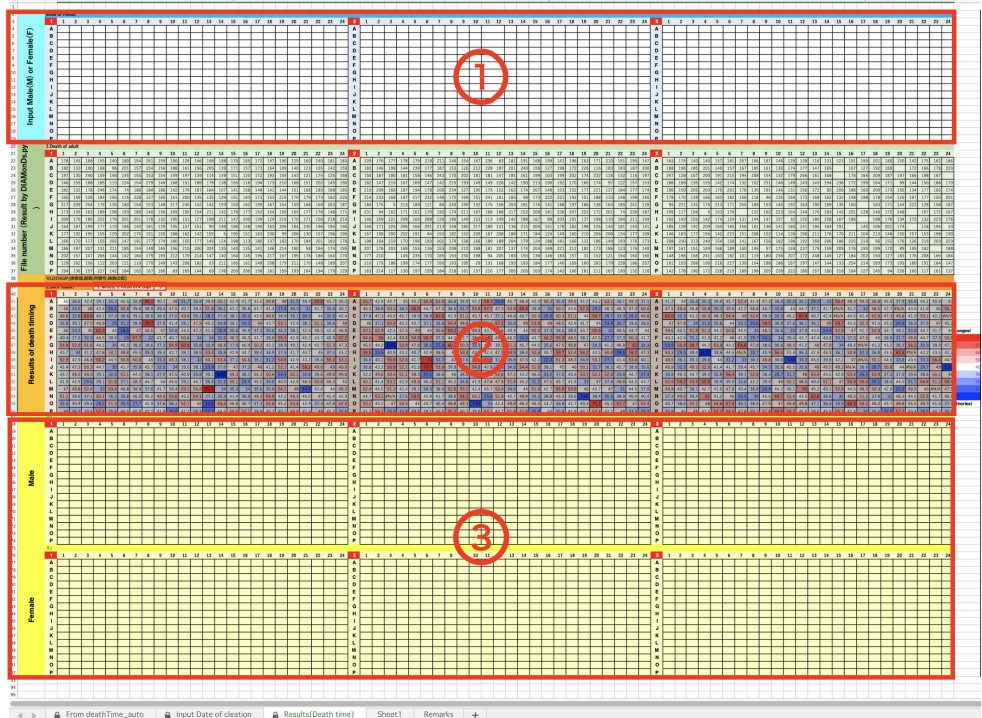

**Appendix 7—figure 3.** DIAMonDS analysis template: tabs for results. (**a**) Input Sex ('M' or"F') of each well into ①. (**b**) The Result is indicated in ②. (**c**) In ③, Results are separated two groups, male and female.

# Appendix 8

## Life-event detection algorithm and software: Sapphire

Present system provides automatic high-accurate life-event detection algorithm including the image processing and signal processing shown in *Figure 1c*. Following text describes quantitative summary and robustness of the algorithm in various parameter spaces.

### 8.1. Data augmentation for network training

Present system performed the semantic segmentation through the deep neural network whose architecture is described in *Figure 1—figure supplement 2b*. Generally, sufficient number of annotation data leads to more accurate inference in deep learning. Two training datasets were prepared for the training of neural network. Both of training datasets include original images obtained by DIA-MonDS at different conditions, and binary images labeled on animal-body by hand. First training dataset has original and labeled images of adults, and it was applied to the training of network for death-detection. Second training datsaset consists of adult- and larva-images and their binary labels. Larva images in the training data were employed to the traning of network for detection of pupariation. Not only adult but also larva images as distractors were applied to the training of the network for eclosion (*Appendix 8—table 1*). The number of training data was magnified by general data augmentation technique ('*Image processing*' in main text).

**Appendix 8—table 1.** Data augmentation for training of FCN designed for animal body segmentation.
Note that the system was trained by only two datasets.

|  | Training data | Training data 2 |  |
| --- | --- | --- | --- |
| **Target event** | **Death** | **Pupariation/Eclosion** |  |
| Target animal in segmentation | Adult | Adult | Larva |
| No. of annotation data | 178 adults | 300 adults | No adult |
|  |  | 5760 larva (distractor) | 1839 larva |
| No. of augmented data | 71,200 adult | 60,000 adults | 183,900 larva |
|  |  | 57,600 larva (distractor) |  |

### 8.2. Animal body segmentation

Present system could detect rough outline of animal body (*Appendix 8-figure 1*).

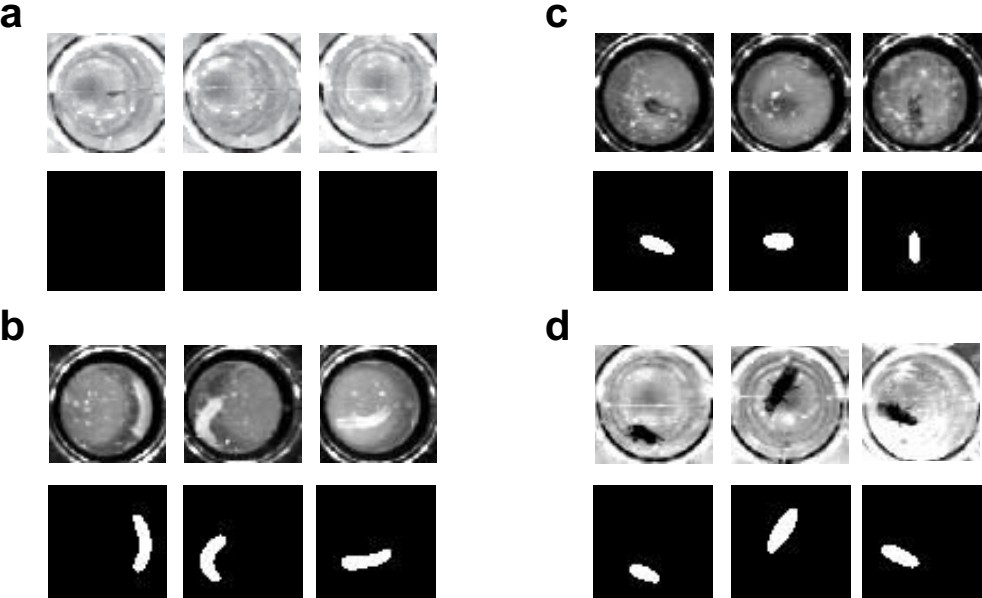

**Appendix 8—figure 1.** Example of segmentation for pupa (**a**), larva (**b**), adult in dataset 1 (**c**), and dataset 2 (**d**). Note that the segmentation correctly captured the animal body in the images obtained by different imaging condition (**c and d**), and correctly 'ignored' pupa (**a**).

In some cases, segmentation errors still appeared. The errors in segmentation could be improved by the increase of annotation data for training of network. In other word, present system showed high performance even though datasets shown in the manuscript were obtained through several imaging condition as well as the number of annotation data was small. In consistent imaging condition (same shape and size of well, plate number, and so on), the system will robustly detect entire life-event. Additional annotation data for every experiment might be also effective in order to promote network training. In other word, present algorithm could reach sufficient accuracy if an appropriate signal processing was applied on the image data even with rough segmentation.

## 8.3. Robustness of detection for CF parameters

In the present system, subtraction between consecutive segmentation images were converted to ChangeFinder (CF) signal which is one of change point detection algorithms1 (see Materials and methods).

For the death detection, the system calculates the CF signal from adult body segmentation (*Appendix 8—figure 2a*). For the pupariation and eclosion detections, two CF signals were calculated from adult and larva segmentations, respectively (*Appendix 8—figure 2b*).

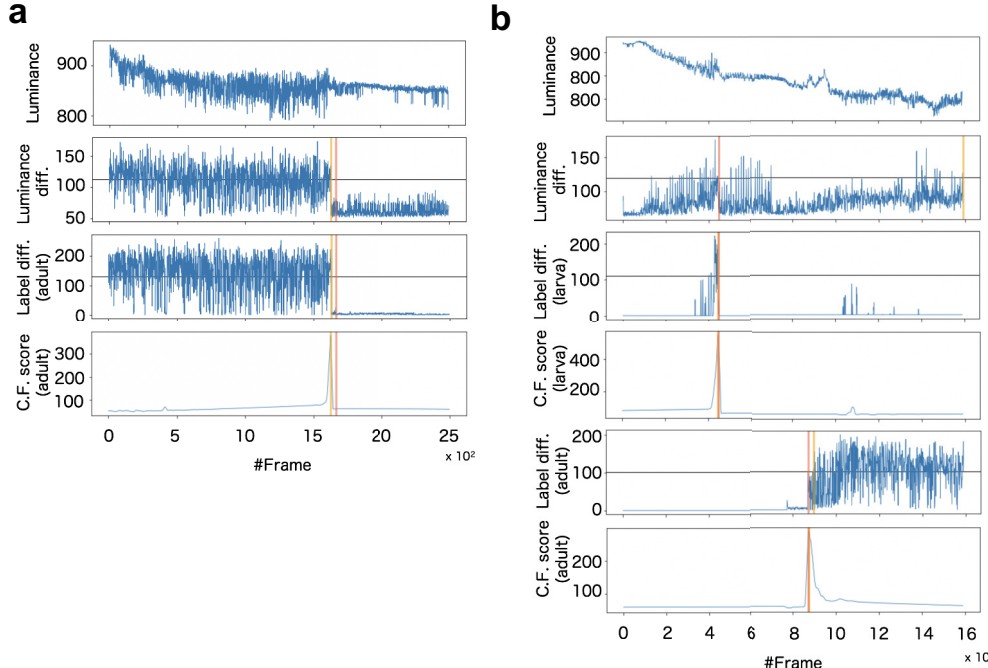

**Appendix 8—figure 2.** Example of signalization from image data and detection of death (**a**) and pupariation-eclosion (**b**). In (**a**), each panel indicates signals obtained as (*top*) luminance at every time steps, (*second panel*) subtraction of luminance between consecutive raw images, (*third panel*) subtraction between consecutive labeled images obtained by larva segmentation, and (*bottom*) ChangeFinder signal calculated by Label diff. Black horizontal lines indicate thresholds for detection of event timing. Red vertical line indicates the manually-detected event timings (pupariation), and yellow vertical lines are automatically-detected event timings (see *supplementary text*). In (**b**), almost same organization with (**a**) except both signals of larva and adult are included for detection of pupariation and eclosion.

In the present system, the timing of life-event transitions is determined as the maximum point of corresponding CF scores ('Max-CF').

To evaluate robustness of detection, we compared several detection methods. Most simple way is just thresholding of the activity signal without change point detection such as CF. We calculated two types of signal from difference between consecutive images. One is the luminance difference ('Luminance diff.') obtained by subtraction between raw images, and another one is the subtraction between segmentation images before the CF ('Seg. diff.'). Threshold for event timing were calculated as total average of the signals ('auto') and manual tuning ('manual'). In comparison of these event detection methods, max-CF exhibited high accuracy in almost all datasets (*Appendix 8—table 2*).

**Appendix 8—table 2.** Summary of consistency between automatic and manual detections of pupariation (a), eclosion (b), and death (c).

The consistency was evaluated as the ratio of individuals whose detected frame differences between the algorithm and human expert were less than 5% for entire frame. Columns indicates different dataset and rows corresponds different detection method for comparison. Consistencies were described as the ratio and the number. The cells with >= 90% consistency were labeled by two asterisks, and one asterisk corresponds the cells equal or greater than 80%, and less than 90%. Note that the total numbers of animal were different depending on dataset because of the elimination of some individuals that could not had the events.

**Pupariation**

|  | Dataset 1 (allevent) | Dataset 2 (160416) | Dataset 3 (180223) |
|---|---|---|---|
| Luminance diff. thre (auto) | 36.6% (102/279) | 0.0% (0/255) | 37.4% (92/246) |
| Luminance diff. thre (manual) | 78.5% (219/279) | 39.6% (101/255) | 39.0% (96/246) |
| Segmentation diff. thre (auto) | 96.4%** (269/279) | 82.4%* (210/255) | 94.7%** (233/246) |
| Segmentation diff. thre (manual) | 96.8%** (270/279) | 91.8%** (234/255) | 96.3%** (237/246) |
| Maximum point of CF signal | 97.5%** (271/279) | 96.1%** (245/255) | 92.3%** (227/246) |

**Eclosion**

|  | Dataset 1 (allevent) | Dataset 2 (160416) | Dataset 3 (180223) |
|---|---|---|---|
| Luminance diff. thre (auto) | 14.6% (34/233) | 0.0% (0/241) | 98.8%** (161/163) |
| Luminance diff. thre (manual) | 60.1% (140/233) | 2.9% (7/241) | 98.8%** (161/163) |
| Segmentation diff. thre (auto) | 82.0%* (191/233) | 95.9%** (231/241) | 99.4%** (162/163) |
| Segmentation diff. thre (manual) | 85.4%** (199/233) | 97.1%** (234/241) | 99.4%** (162/163) |
| Maximum point of CF signal | 86.3%* (201/233) | 97.5%** (235/241) | 99.4%** (162/163) |

**Death**

|  | Dataset 1 (150625) | Dataset 2 (171013) | Dataset 3 (180223) |
|---|---|---|---|
| Luminance diff. thre (auto) | 62.0% (237/382) | 97.9%** (281/287) | 10.3% (14/136) |
| Luminance diff. thre (manual) | 94.0%** (359/382) | 97.9%** (281/287) | 10.3% (14/136) |
| Segmentation diff. thre (auto) | 76.4% (292/382) | 97.2%** (279/287) | 99.3%** (135/136) |
| Segmentation diff. thre (manual) | 93.5%** (357/382) | 97.6%** (280/287) | 100.0%** (136/136) |
| Maximum point of CF signal | 94.8%** (362/382) | 93.4%** (268/287) | 98.5%** (134/136) |

CF has two parameters, $r$ and T. The $r$ is a discounting rate and $T$ is the length of a time window for averaging. We examined in robustness of detection accuracy for these two parameters in max-CF. For all event target and all datasets, max-CF demonstrated high-accuracy in wide range of the parameter, even the accuracy is relatively sensitive for discounting $r$ than smoothing $T$ (*Appendix 8—figure 3–5*).

# Pupariation

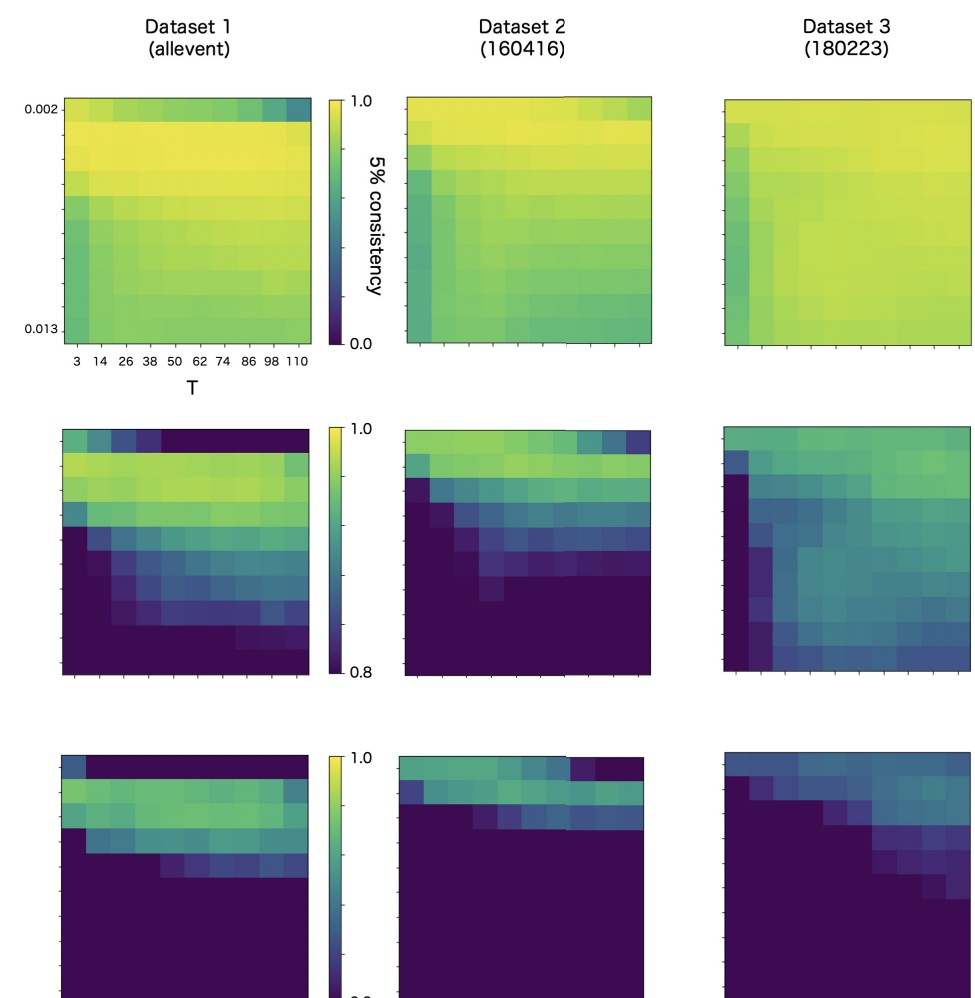

**Appendix 8—figure 3.** Robustness of automatic detection system for ChangeFinder parameters at pupariation detection. Each panel shows 5% consistency as heatmap for *T* (x-axis) and *r* (y-axis) of ChangeFinder's parameter. Panels are placed for different datasets (column) and color-scales (row).

Present system used *r* = 0.003 except death detection in the datasets 1 (150626). In the data, *r* = 0.009 because the datasets has significantly few images.

## 8.4. Application of the algorithm

The system captures the animal body and its trajectory. Therefore, one could estimate behavioral history and total activity of individual *Drosophila* by displaying of trajectory and its summation in a well. (*Video 1*).

In addition, the system could demonstrate dynamic visualization of life-event transitions of *Drosophila* population (*Appendix 8—video 1*).

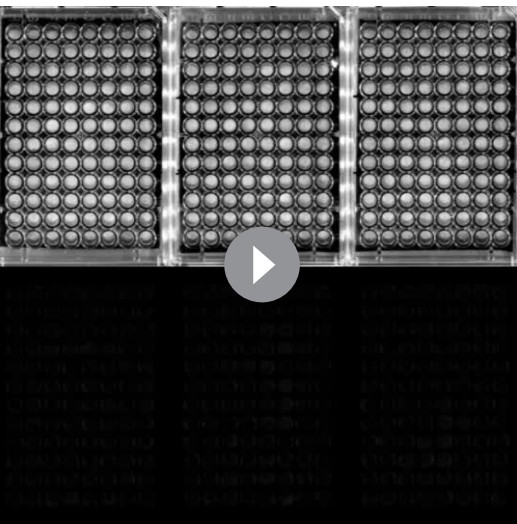

**Appendix 8—video 1.** Example of sequential detection of pupariation and eclosion by Sapphire.
https://elifesciences.org/articles/58630#A8video1

Through the visualization, the users could easily recognize timing differences between populations they were bled on different plates.

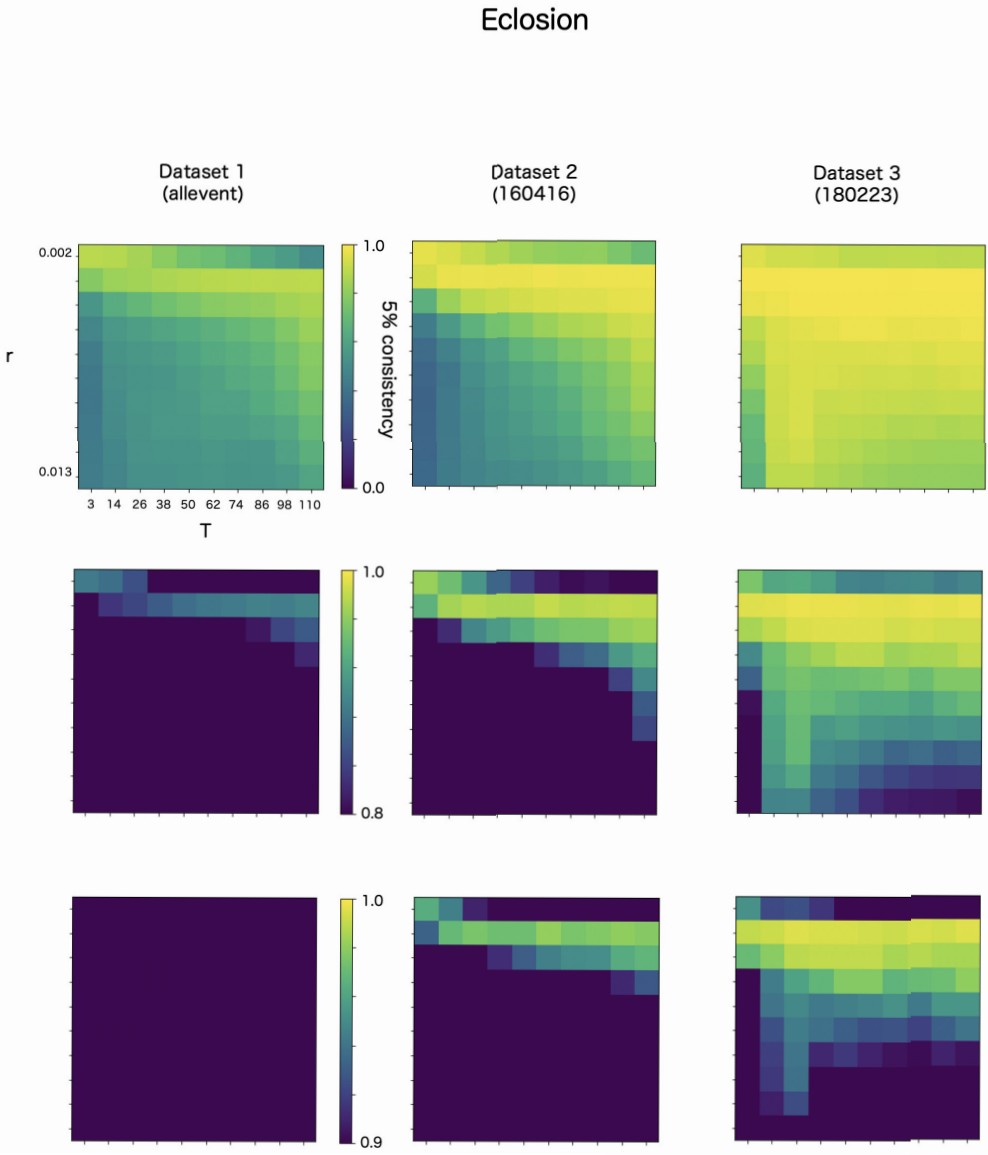

**Appendix 8—figure 4.** Robustness of automatic detection system for ChangeFinder parameters at eclosion detection. Each panel shows 5% consistency as heatmap for *T* (x-axis) and *r* (y-axis) of ChangeFinder's parameter. Panels are placed for different datasets (column) and color-scales (row).

# Death

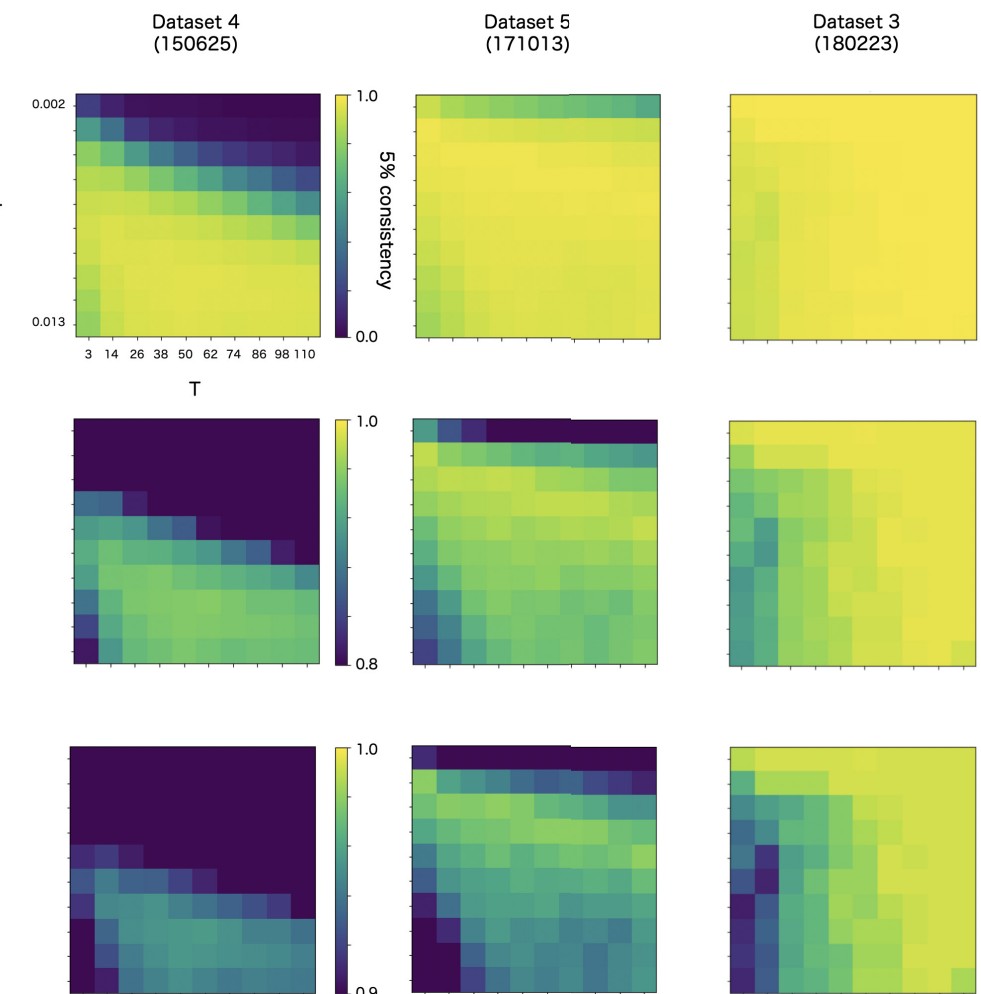

**Appendix 8—figure 5.** Robustness of automatic detection system for ChangeFinder parameters at death detection. Each panel shows 5% consistency as heatmap for $T$ (x-axis) and $r$ (y-axis) of ChangeFinder's parameter. Panels are placed for different datasets (column) and color-scales (row). Most right column (dataset 3, 180223) is same dataset in the *Appendix 8—figures 3* and *4*.

