## [Decision Letter]

**Acceptance summary:**

We believe that the monitoring assay that the authors here developed will be useful in many fly labs and anticipate that this work will impact the research of very different types of fly research.

**Decision letter after peer review:**

Thank you for submitting your article "The *Drosophila* Individual Activity Monitoring and Detection System (DIAMonDS)" for consideration by *eLife*. Your article has been reviewed by two peer reviewers, and the evaluation has been overseen by a Reviewing Editor and K VijayRaghavan as the Senior Editor. The following individuals involved in review of your submission have agreed to reveal their identity: Jonathan Andrews (Reviewer #1); Michael B O'Connor (Reviewer #2).

The reviewers have discussed the reviews with one another and the Reviewing Editor has drafted this decision to help you prepare a revised submission.

We encourage you to carefully read the critiques and address all the issues that are listed below.

Reviewer #1:

The paper entitled ‘The *Drosophila* Individual Activity Monitoring and Detection System (DIAMonDS)’ highlights a new detection/tracking system which utilizes a flatbed CCD scanner to track and identify multiple life cycle events (pupariation, eclosion, and death) using a newly developed algorithm. In support of this novel monitoring system, the authors provide multiple examples of the tracking system in action, including analysis of larval and adult movement and the detection of pupariation and eclosion at a high temporal resolution. The authors also provide several examples of more complex experiments which can be accomplished in a high-throughput manner using DIAMonDS, including lifespan and stress resistance assays. As described, this system would provide a researcher with an automated tool for measuring the timing of multiple major developmental milestones in *Drosophila* development – essentially allowing for more accurate and less labor intensive observation.

While DIAMonDS is certainly valuable in its current incarnation, the authors do note a number of worrying limitations which I believe should be resolved prior to publication, and there are several areas of the manuscript where I believe more detail is warranted.

1) As DIAMonDS detects changes between the static and active phases in the *Drosophila* lifecycle through changes in motion, it is essential that the authors demonstrate (or provide an explanation of) how they discriminate between less motile stages of development (or death) and normal cessation of motion while alive, such as in grooming or sleep behavior in the adult.

2) Similar to critique #1, this type of motion detection may not be as effective in animals with some form of locomotor defect. An additional experiment demonstrating that DIAMonDS can reliably detect and classify larvae or adults with reduced locomotion is prudent to demonstrate that it can work, even if the flies are impaired in some way.

3) It is currently unclear how the DIAMonDS system handles events that occur "off-camera", as can be observed in frame 77-79 of Video 1. This may be a potential sources of error during tracking – for example, if a larvae crawls into an area where it is not observed and pupates, or if an animal dies out of view.

4) It is unclear how (or if) the system would discriminate between pupation and a dead larvae. A failure to account for this could easily result in a false-positive for pupation.

5) Paragraph four subsection “Pupariation and eclosion timing detection by DIAMonDS”: I was unable to locate a description of the semi-automatic (TH) methods presented here in the Materials and methods section.

6) The inability of the methods described in the manuscript to handle *Oregon-R* or *Canton-S* strains significantly limits the usefulness of the system. A set of optimal conditions for common laboratory strains should be included with the manuscript.

7) As admitted by the authors, the size of the wells may adversely affect fly health. It would be worthwhile to see a comparison between the smaller chambers and a larger chamber, so as to allow for future users of the system to make more informed decisions about how to implement it.

Reviewer #2:

The manuscript by Seong et al. describes the development of a sophisticated activity monitoring system that is able to determine with, great accuracy, the timing of major life stage transitions during *Drosophila* development. Specifically, the system relies on time lapse imaging and A.I. based learning to pinpoint three transitions: (1) larval to pupal, (2) pupal to adult, (3) adult to death. The basic principle is to establish the location of a larva, pupa or adult at each time point within either a 96 or 384 well plate and then determine if it has changed at the next time point. Since larva and adults are motile and pupa and dead flies are not then it is conceptually easy to distinguish the stage transition through location changes by monitoring location changes. The authors demonstrate that the system works, at least for the *W1118* genetic background, and is able to replicate known developmental characteristics such as the fact the females typically eclose a few hours before males and that the timing of the larval to pupal transition is diet (sugar) sensitive. They also demonstrate that it can be used to establish Kaplan-Meier lifespan curves which are capable of distinguishing environmental effects on adult lifespan such as the presence of DDT or paraquat in the food. Overall this system appears to have great potential for quantitatively measuring a number of developmental parameters that are presently very tedious to determine manually and are therefore not amenable to high throughput procedures that are needed for genetic and drug screening.

I do not feel competent to comment on the software development and AI procedures used to train the system other than to say that they appear to work quite reliably as long as the optics are not disturbed. Herein lies the biggest disappointment.

1) The authors conclude their Results section by saying that they cannot reliably measure lifespan in common strains such as *Oregon-R* and *Canton-S* because of accidental death effects due to such issues as water condensation in the wells and also due to blockage of the optical light path by the spread of food particles and feces on the well lid that obscures detection of the fly's position during imaging. The authors say that additional refinements of the system will be needed to overcome these challenges for adult lifespan analysis. I wonder, however, if the authors have tried something as simple as replacing the lid of the microtiter dish at some frequency during the lifespan measurements. I recognize that the entire chamber will need to be immersed in a CO_2_ chamber or cooled to knock the flies out and that this may influence the lifespan kinetics, but have the authors attempted anything like this as a work around to the degenerating light path and water accumulation issues during aging studies?

Despite this drawback, I think the system still has significant utility for assaying environmental and genetic effects on larval to pupa and pupal to adult transitions and this makes it is worth communicating to the *Drosophila* research community.

---

## [Author Response]

Reviewer #1:[…]1) As DIAMonDS detects changes between the static and active phases in the *Drosophila* lifecycle through changes in motion, it is essential that the authors demonstrate (or provide an explanation of) how they discriminate between less motile stages of development (or death) and normal cessation of motion while alive, such as in grooming or sleep behavior in the adult.

Thank you for the reviewer’s critical critique. After the segmentation by deep neural network, Sapphire obtained the ChangeFinder (CF) signal which characterized change points. The CF signal includes both long- and short-time scale factors and a change point is determined as the maximum point showing significantly distinct change and the longest stationary state after the change. Therefore, transient changes from dynamic to static state such as sleep could be effectively ignored by the process because the CF signal is often much smaller than the one of fatal cessation like a death and pupariation (please see Author response image 1).

The time scale parameters of CF calculation were optimized for the detection depending on the total number and interval between frames. However, robustness of detection was guaranteed in wide range of the parameters and it does not require fine-tuning (see Appendix 8—figure 3-5).

**Author response image 1. sa2fig1:** Conceptual figure of activity signal (upper panel) and ChangeFinder (CF) signal (lower panel). Temporal cessation of *Drosophila* behaviour like a sleep showed relatively low CF score than the long-time cessation such as death. So, the present system could correctly capture the event as a maximum point of CF score. NB. All the signals were idealized for understanding.

2) Similar to critique #1, this type of motion detection may not be as effective in animals with some form of locomotor defect. An additional experiment demonstrating that DIAMonDS can reliably detect and classify larvae or adults with reduced locomotion is prudent to demonstrate that it can work, even if the flies are impaired in some way.

As the reviewer mentioned, rich locomotor activity is desirable for robust detection in the present system. However, the present algorithm can determine fatal cessation even the larval locomotion during alive is less motile because the CF signal of fatal cessation is typically higher than the ones of insufficient locomotor activity (Author response image 2). Of course, the system could not detect the individual which shows extremely low signal or completely loss the event, that is, the target event did not occur due to fly impairment.

As treatment for the cases, the system could make users rapidly identify and access such the individuals visually by monitoring consequent images of animal body segmentation. Furthermore, the flies without target event could be easily eliminated, for example, the individual that could not reach the eclosion after pupariation could be excluded from survival ratio of adult *Drosophila* by just editing csv file named as “Blacklists” in Sapphire (see Appendix 7).

To demonstrate the performance of the DIAMonDS in the case of hypoactive flies, we used decapitated females which show stationary status throughout their life. DIAMonDS effectively detected their death time points that were comparable to the detection by visual handling, indicating that the DIAMonDS shows good performance even in such low active situations. We have added the data in Figure 4—figure supplement 3 and the following description in the main text.

“Next, we tried to test whether the DIAMonDS can effectively detect the death time points even in flies showed a very reduced amount of activity. As a hypoactive fly model, we used decapitated females who keep motor skills but hardly moved until their death (Ejima and Griffith, 2008). DIAMonDS showed relatively good results that were comparable to the visual results although there was a tendency for the accuracy to reduce slightly in comparison with the case of wild-type flies (Figure 4—figure supplement 3).”

**Author response image 2. sa2fig2:** Conceptual figure of high and low activity. Upper panels indicates activity signal and lower plans are CF values. In low activity animal, both signal shows relatively low intensity, however, the maximum point still indicates continuous cessation which corresponds to death. NB. All the signals were idealised for understanding.

3) It is currently unclear how the DIAMonDS system handles events that occur "off-camera", as can be observed in frame 77-79 of Video 1. This may be a potential sources of error during tracking – for example, if a larvae crawls into an area where it is not observed and pupates, or if an animal dies out of view.

Invisibility including off-camera as far as it is temporal could be ignored in the system by the same reason as the answers for critique 1 and 2. However, as reviewer #1 mentioned, in some cases, if the event, such as pupariation and death, occurs out-of-camera-frame, there is a possibility that the accuracy of detection will slightly affect. In order to reduce the errors, a shorter time-lapse interval and adjusting the amount of food in the wells to eliminate blind spots could be effective. Therefore, we have added the following description in the Discussion.

"Fifth, occasionally, an animal cannot be detected in the image by the existence of a blind area of scanning. The system is not greatly affected even if "off-camera" occurs during an active state of an animal. But the accuracy of event detection might be slightly affected when "off-camera" occurs just at the timing of the event shift. It could be reduced by increasing the spatial- or temporal-resolution to eliminate the blind-area of each well."

4) It is unclear how (or if) the system would discriminate between pupation and a dead larvae. A failure to account for this could easily result in a false-positive for pupation.

As the reviewer mentioned, the system could not discriminate between pupariation and dead larvae when it monitored only pupariation at a glance. If the system continued monitoring after pupariation, the system could discriminate by the referring of CF signal corresponding to “eclosion” because it is a proof of the larva was not dead. However, dead larvae can easily be discriminate after finishing time-lapse scanning and could be manually excluded by making the “Blacklists” file in Sapphire analysis.

We would like to add the following description in Materials and methods section:

“After time-lapse scanning, we can easily discriminate “out-of-event”, such as larval, and, pupal dead individuals. Sapphire can exclude “out-of-event” wells by making the “Blacklists” file (Appendix 7).”

5) Paragraph four subsection “Pupariation and eclosion timing detection by DIAMonDS”: I was unable to locate a description of the semi-automatic (TH) methods presented here in the Materials and methods section.

We apologize our incomplete information and incorrect reference for semiautomatic (TH) method which was explained not in “Materials and methods” but in “Appendix 8”.

We rewrote the sentence in the main text as follows:

“Arbitral image sequences such as raw images and the segmentation images could be analyzed in Sapphire (see Appendix 8). Sapphire calculates the CF signal obtained from the subtraction of consecutive segmentation images as fully automated method (CF method). Sapphire determined the timing of life-event by applying just thresholding on the arbitral time series data specified by user (TH method). At first, threshold is automatically calculated as an average of maximum and minimum value of the signal (auto-TH) and the threshold is modifiable by hands depending on user’s demand (manual-TH). Accuracy of CF method and auto- and manual-TH method on raw image subtraction was quantified in (Appendix 8).”

6) The inability of the methods described in the manuscript to handle Oregon-R or Canton-S strains significantly limits the usefulness of the system. A set of optimal conditions for common laboratory strains should be included with the manuscript.

We appreciate this comment. To find optimal conditions for common laboratory strains, we tried to measure the survival of *Oregon-R*, *Canton-S*, and *w1118* strains in different well-size and plate replacement cycle. We observed that the survival of flies was improved by replacing the microplate every 2-3 days per week. On the other hand, the bigger well-size did not improve the viability.

We would like to add the results as Figure 6—figure supplement 1 and describe as follows in the main text:

“To find common optimum conditions for measuring the lifespan of other laboratory strains, we have tried to observe the viability of three strains (*w1118*, *Oregon-R,* and *Canton-S*) in condition with several microplate well-size (96-well, 48-well, and 24-well) (Figure 6—figure supplement 1). The viabilities were measured at the time point at 1st, 2nd, 3rd, and 4th weeks with once plate replacement a week. We observed that the reduced viability was not rescued by changing to bigger microplate-well size, and somewhat, increasing well-size might have harmful effects on the survival of aged flies (Figure 6—figure supplement 1A-C). We next tested the two different plate replacement cycles (once a week, or once 2-3 days) using *Oregon-R* and *Canton-S* in 96-well microplate. We observed a shorter plate replacement cycle could effectively improve the viability (Figure 6—figure supplement 1B,C). However, because plate replacement is required a lot of work time, it seems to be not suitable for the high-throughput experiments. Further optimization of the experimental conditions will be necessary to effectively use DIAMonDS to conduct lifespan tests on a wide range of fly strains.”

7) As admitted by the authors, the size of the wells may adversely affect fly health. It would be worthwhile to see a comparison between the smaller chambers and a larger chamber, so as to allow for future users of the system to make more informed decisions about how to implement it.

We appreciate this comment. We tried to use different size of microplates (96well, 48-well, and 24-well) for measuring the pupariation and eclosion. We observed that timings of pupariation and eclosion were slightly but significantly shorten both in the 48-well and 24-well conditions and the pupal duration might have the fewer effects of well-size (Figure 2—figure supplement 5).

We would like to add the results as Figure 2—figure supplement 5 and describe as follows in the main text:

“To understand the effect of chamber size on fly development, we used three different sizes of microplates (96-well, 48-well, and 24-well) contained normal fly media for measuring pupariation and eclosion. We observed that timings of pupariation and eclosion were slightly but significantly shorten both in the 48well and 24-well conditions and the pupal duration might have the fewer effects of well-size, suggesting that the chamber size might affect fly’s development (Figure 2—figure supplement 5).”

Reviewer #2:[…]1) The authors conclude their Results section by saying that they cannot reliably measure lifespan in common strains such as Oregon-R and Canton-S because of accidental death effects due to such issues as water condensation in the wells and also due to blockage of the optical light path by the spread of food particles and feces on the well lid that obscures detection of the fly's position during imaging. The authors say that additional refinements of the system will be needed to overcome these challenges for adult lifespan analysis. I wonder, however, if the authors have tried something as simple as replacing the lid of the microtiter dish at some frequency during the lifespan measurements. I recognize that the entire chamber will need to be immersed in a CO_2_ chamber or cooled to knock the flies out and that this may influence the lifespan kinetics, but have the authors attempted anything like this as a work around to the degenerating light path and water accumulation issues during aging studies?

We appreciate this supportive comment. As mentioned in reply to the point #7 of reviewer #1, to find optimum condition for lifespan analysis, we tested a well-size effect on survivorship of adult fly strains. The result indicated that adult survival did not improve increasing the well-size of the microplate and rather reduced in *Oregon-R* and *Canton-S* strain at 3rd and 4th weeks. We also tried to measure the percent survival of *Oregon-R* and *Canton-S* strains in different plate replacement cycle conditions. We observed that the survival of flies was significantly improved by replacing the microplate every 2-3 days per week in comparison with the once a week replacing cycle. We added the new figures and sentences as described in reply to the pojnt #7 of reviewer #1.

As mentioned by reviewer #2, anesthetizing such as CO_2_ and cooling the body might influence the lifespan. Therefore, we did not use any anesthetizing treatment for lifespan experiments. We used a fly aspirator and transferred the living flies directly in the wells of the microplate. This is mentioned at Appendix 3.